# UpsFrac v1.0: An open-source software for integrating modelling and upscaling permeability for fractured porous rocks

Tao Chen[1,2], Honghao Sheng[1,2], Yu Zhang[1,2], Fengxin Kang[1,2,3]

[1]College of Earth Science and Engineering, Shandong University of Science and Technology, No. 579 Qianwangang Road, Qingdao, 266590, China
[2]Shandong Key Laboratory of Geothermal Clean Energy, 272000 Jining, China
[3]Shandong Provincial Bureau of Geology & Mineral Resources (SPBGM), No. 74 Lishan Road, Jinan, 250013, China

*Correspondence to*: Tao Chen (chentao9330@gmail.com)

**Abstract.** The efficient and accurate simulation of fluid flow and heat transport underground plays an important role in groundwater migration and geothermal resource prediction. Rock fractures are complex in geological settings, exhibiting multiple-scale properties with varied patterns under different geological conditions. Modelling and upscaling the permeability of fractured porous rocks are both important and sophisticated processes in numerical simulations. While existing tools like MRST offer upscaling capabilities, they often lack integrated workflows for automated uncertainty quantification and direct coupling between fractal-based DFM generation and flow-based upscaling within a single framework. In this study, we propose an integrated methodology for modelling and upscaling the permeability of fractured porous rocks and develop an open-source software, UpsFrac, to implement this approach. The software considers complex fracture geometries in discrete fracture models (DFM) that are created deterministically and stochastically. UpsFrac employs conforming grid-based DFM with explicit matrix-fracture coupling, ensuring higher accuracy for complex fracture networks. The software can characterize the complexity of fractured porous rocks, including power law (fractal) length distribution, correlations between fracture length and aperture, and the effects of rock matrix properties. The state-of-the-art upscaling method, the multiple-boundary fracture upscaling (MFU) method, is applied to calculate equivalent fracture permeability, which links the fine-scale discrete fracture model to the coarse-scale equivalent fracture model. The current implementation is 2D and MATLAB-based, built upon fracture modelling code ADFNE and reservoir simulation code MRST, which can easily run DFM ensembles for uncertainty analysis. The code is available in open repositories to encourage modelling and upscaling of complex fractured porous rocks, allowing users to develop their own routines within the current framework and benefiting a broader community.

## 1 Introduction

Fractures exist widely in geological settings and are caused mainly by changes in geomechanical stress, chemical erosion, and thermal stress (Gu et al., 2020; Gudmundsson, 2011; McDermott and Kolditz, 2006; Molnar et al., 2007; Zoback et al., 2003). These fractures can significantly influence the movement of fluids through rock formations, affecting groundwater

flow, contaminant transport, and the overall stability of geological formations (Cook et al., 2005; Marinos and Carter, 2018; Myers, 2012; Odling and Roden, 1997). In geothermal or petroleum reservoirs, fractures enhance the permeability and porosity, efficiently extracting heat and hydrocarbons (Ghassemi, 2012; Gong et al., 2021). Understanding the characteristics and behaviours of fractures is essential for reducing environmental risk, optimizing energy production, and implementing proper management strategies.

Quantitative analysis of reservoirs is essential for predicting sustainable production in geothermal or hydrocarbon systems, which mainly involves data collection, fractured porous rock modelling, upscaling, numerical simulation, and history matching (Andrews et al., 2019; HosseiniMehr et al., 2022; Liu and Reynolds, n.d.; Mejia et al., 2021; Neuman, 2005; Viswanathan et al., 2022). Discrete fractures occur at multiple scales, as illustrated in Fig. 1. Characterizing fractured porous media mainly involves gathering geological, hydrological, and geophysical data to obtain fracture and matrix properties within the reservoir, e.g., fracture orientation, length, density, and connectivity. Once the conceptual model of the fractured porous rock is built based on characterization, there are two primary numerical simulation approaches used to represent these fractures (Berre et al., 2018): the Discrete Fracture Model (DFM) or Discrete Fracture Network (DFN) and the Equivalent Fracture Model (EFM). The DFM approach explicitly represents each fracture within the reservoir, capturing the complexities of fluid flow and coupled processes. The EFM simplifies the representation by averaging the effects of fractures, making it computationally more efficient while still providing valuable insights. Upscaling techniques are required to calculate equivalent properties and construct a computationally efficient EFM based on the DFM. Accurate upscaling is particularly important for studying the heterogeneity and anisotropy of the reservoir and the existence of representative elementary volume (Wang et al., 2023), allowing for a better understanding of how these factors influence geothermal production (Gottron and Henk, 2021; Rajeh et al., 2019; Renard and Ababou, 2022). Modelling and upscaling discrete fracture models are prerequisites for the following numerical simulations and history-matching procedures.

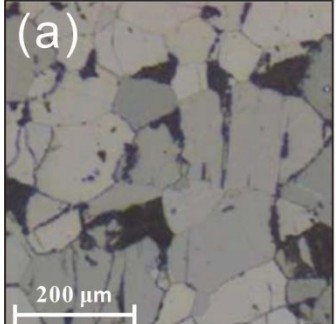 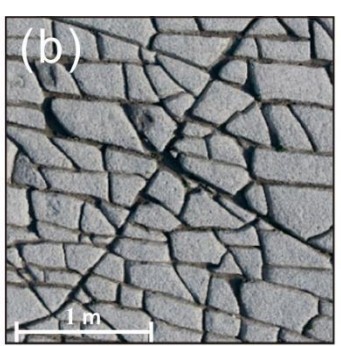 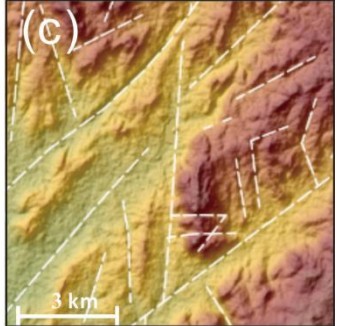

**Figure 1: Multiple-scale fractures in geological settings. Adapted from Cheng and Wong (2018), Palamakumbura et al. (2020), and Weismüller et al., (2020).**

Fractures are complex in their geometries and introduce uncertainty into models, primarily due to limited data availability and the influence of underground rock and fluid dynamics (MacQuarrie and Mayer, 2005; Srinivasan et al., 2018), which poses significant challenges in accurately characterizing and modelling fractured porous media. Numerous codes have been developed to advance the characterization and modelling of such media to address these issues. Hardebol and Bertotti (2013) developed DigiFract, a software solution that enables comprehensive fracture data collection from outcrops with greater efficiency than conventional surveying methods, enabling faster collection of larger and more accurate fracture datasets for better characterization of fractured rocks. Healy et al. (2017) developed FracPaQ, an open-source MATLAB™ toolbox designed for quantifying fracture patterns from two-dimensional images at multiple scales, such as thin section micrographs, geological maps, outcrop or satellite images. Alghalandis (2017) developed an open-source software ADFNE for stochastic modelling of discrete fracture networks in two- and three-dimensional applications. ADFNE provides a platform for visualizing and analyzing fracture networks, offering a valuable tool in fractured rock mechanics. Welch et al. (2020) developed DFN Generator v2.0, an innovative tool that simulates natural fracture network evolution based on geomechanical principles rather than stochastic methods, enabling more realistic modelling of fracture networks at kilometer scales. Ovaskainen (2023) developed the Python package fractopo, which provides tools for data validation and analysis of fracture trace observations digitized from base maps like drone images of outcrops or digital elevation models.

More recently, Borghini et al. (2024) introduced the Fracture Analyser, a simple and user-friendly Python tool, for the 2D analysis of fracture patterns in rock outcrops. The tool quantifies number, length, orientation, position, fracture density, etc., providing an efficient, flexible and accurate tool for the characterization and modelling of natural fractures. While several sophisticated tools exist for fractured porous rock modelling—such as PorePy (Keilegavlen et al., 2021) for discrete fracture-matrix simulations, DFNWorks (Hyman et al., 2015) for DFN flow and transport, SHEMAT-Suite (Keller et al., 2020) for coupled flow and heat transport, and OpenGeoSys (Kolditz et al., 2012) with its DFM modules, there remains a significant gap in linking such fracture geometric properties to the equivalent properties of the equivalent fracture model when doing numerical simulation. Few of these tools provide an integrated workflow that directly couples fractal DFM generation with flow-based permeability upscaling within a single framework. MRST (Wong et al., 2021) offers upscaling capabilities through embedded discrete fracture models (EDFM), but this approach uses non-conforming grids that may result in reduced accuracy for highly connected fracture networks. To the best of our knowledge, no open-source platform currently bridges fractal theory with practical permeability upscaling while enabling automated uncertainty quantification through ensemble-based analysis.

UpsFrac advances the state-of-the-art by providing one of the first integrated open-source solution bridging fractal theory and practical permeability upscaling. The platform unifies four key components: (1) fractal DFN generation combining deterministic and stochastic models with power-law distributions (Corral and González, 2019) and physically-based aperture-length correlations, capturing multi-scale characteristics of fractured systems; (2) direct coupling architecture

integrating DFM modelling with MFU (Chen et al., 2015) for permeability upscaling without file conversions, employing TPFA/MPFA schemes (Sandve et al., 2012) on conforming grids; (3) automated uncertainty quantification enabling efficient Monte Carlo analysis by streamlining workflows from fracture realization to equivalent properties calculation; and (4) an extensible MATLAB framework with modular architecture for incorporating new algorithms. This integrated approach streamlines uncertainty quantification, making large-scale stochastic analysis accessible to the research community. Table 1 positions UpsFrac relative to existing tools, highlighting its unique integrated workflow capability.

**Table 1: Comparison of UpsFrac with existing fractured porous rocks modelling tools***

| Feature | UpsFrac | PorePy | DFNWorks | MRST | OpenGeoSys |
|---|---|---|---|---|---|
| **Programming Language** | MATLAB | Python | C++/Python | MATLAB | C++ |
| **Open Source** | Yes | Yes | Yes | Yes | Yes |
| **Fracture modelling** | | | | | |
| Stochastic DFN | Yes | Limited[1] | Yes | Yes | No |
| Power-law distribution | Yes | Limited[1] | Yes | Yes | No |
| Aperture-length correlation | Yes | Limited[1] | Yes | Yes | No |
| **Flow Simulation** | | | | | |
| DFM flow scheme | TPFA/MPFA | MPFA/TPFA/ MFD | FEM/TPFA | TPFA/MPFA | FEM |
| Matrix-fracture coupling | Yes | Yes | Yes | Yes | Yes |
| **Fracture Upscaling** | | | | | |
| Permeability upscaling | Yes | No | No | Yes | No |
| Integrated DFM-to-EFM workflow | Yes | No | No | Limited[1] | No |
| Automated uncertainty quantification | Yes | No | No | No | No |

*TPFA: Two-Point Flux Approximation; MPFA: Multi-Point Flux Approximation; MFD: Mimetic Finite Difference; FEM: Finite Element Method; DFN: Discrete Fracture Network; DFM: Discrete Fracture Model; EFM: Equivalent Fracture Model.

[1]: Requires manual scripting.

## 2 Methodology and Theoretical Framework

UpsFrac software has three main components: modelling fractured porous rocks, fractured porous media upscaling and equivalent permeability visualization. For the fractured porous rock modelling part, the Matlab code ADFNE (Alghalandis, 2017) is required to supply basic functions for creating discrete fracture networks. For the flow-based upscaling procedure, the Matlab code MRST (Lie, 2019) is applied to solve the flow equations in the discrete fracture model. The detailed workflow of UpsFrac is shown in Fig. 2.

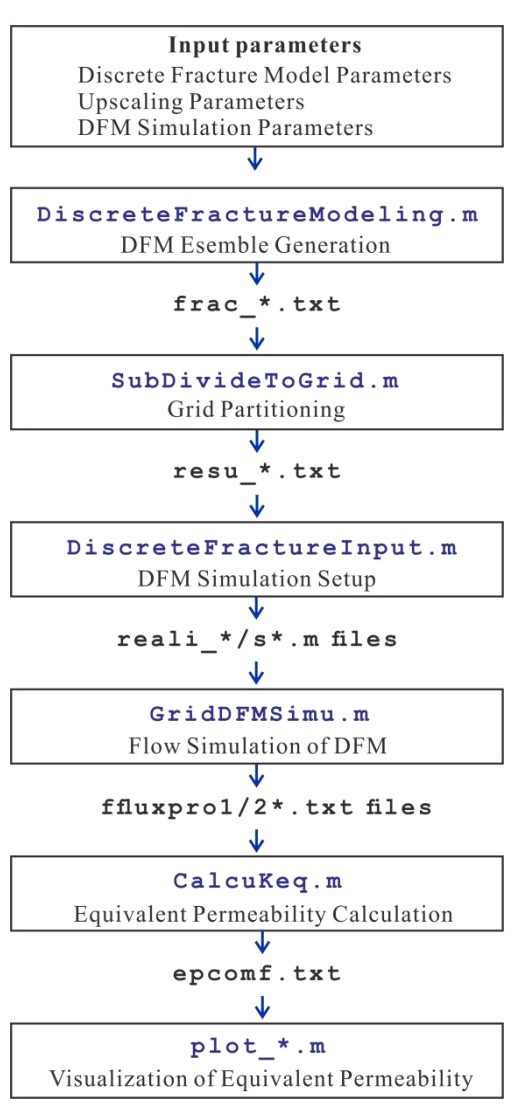

**Figure 2: Implementation workflow of UpsFrac showing script execution sequence and data flow.**

## 2.1 Modelling fractal fractured porous rock

The two-dimensional domain should be defined, e.g., as a rectangle with dimensions Lx and Ly, for modelling fractured porous rocks. The discrete fracture model can consider the discrete fracture network as well as the rock matrix. Fracture modelling includes both deterministic and stochastic approaches. After the fracture geometries are defined, the rest of the domain is filled with the rock matrix. The matrix can be characterized by permeability, porosity, etc.

For deterministic fractures, two endpoints, A(x1, y1) and B(x2, y2), and the fracture aperture should be defined for each fracture. Deterministic fractures are typically used for large-scale features such as faults or fractures with well-constrained geometries, which are important for modelling flow and heat transport.

For stochastic fractures, the discrete fracture network is determined by different geometrical properties (Bonnet et al., 2001) , such as fracture length, aperture, position, and orientation (Xu and Dowd, 2010). The stochastic fracture is mainly based on the software ADFNE. Furthermore, UpsFrac is capable of modelling the truncated power-law length distribution which describes the fractal characteristics of natural fractures. The probability density function for the fracture length l can be written as (Corral and González, 2019; Hyman et al., 2016; Massart et al., 2010):

$$n(l) = \frac{\alpha-1}{l_{min}(1-(\frac{l_{min}}{l_{max}})^{\alpha-1})}(\frac{l_{min}}{l})^{\alpha}, \qquad (1)$$

where $l_{min}$ and $l_{max}$ are the lower and upper bound of the fracture length, $\alpha$, ranging from 1.3 to 3.5, is $\alpha$ power-law exponent influenced by the growth properties of fractures (Bonnet et al., 2001).

Furthermore, the fracture aperture,w, can be correlated to fracture length in UpsFrac by the following power law expression:

$$w = \gamma l^D, \qquad (2)$$

where γ is the coefficient related to mechanical properties of fractured rocks and D denotes the correlation exponent, which signifies the mechanical interaction among closely positioned fractures. D  may vary from 0.5 to 1 resulting from observations in the field. D = 0.5 represents complex open-mode fractures with a constant fracture toughness (Klimczak et al., 2010; Olson, 2003), while D = 1.0 occurs in faults and shear deformation bands under constant driving stress (Vermilye and Scholz, 1995).

Users can use the ADFNE framework to create stochastic realizations for fracture orientation, density, and location. Furthermore, users can develop their models to create stochastic fractures. All the fracture geometric parameters are defined in the modelling framework. The generated fracture geometries are exported in a standardized format that ensures seamless integration with subsequent upscaling procedures.

## 2.2 Upscaling permeability for fractured porous media

The upscaling procedure mainly involves finding the equivalent properties on coarse-scale grid blocks based on the information from fine-scale discrete fracture models. The first step is to divide the fracture domain into a Cartesian grid with nx×ny blocks. Each fracture is geometrically clipped at grid boundaries to identify its intersection segments within individual blocks. The fracture segments within each grid block are defined by their endpoint coordinates A (x1, y1) and B (x2, y2) and aperture values (Fig. 3), providing the geometric and hydraulic parameters necessary for flow simulation. It should be noted that the dimension of the Cartesian grid is (Lx/nx) × (Ly/ny); accordingly, all fracture endpoints are constrained within the individual grid block boundaries.

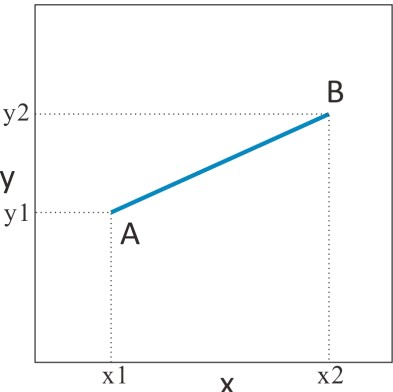

Figure 3: Schematic diagram of a fracture represented by endpoints A and B.

Next, the clipped fractures within each Cartesian grid form local discrete fracture networks for numerical simulation. The TPFA or MPFA schemes (Sandve et al., 2012) are employed to accurately model fluid flow in these highly heterogeneous media. Each grid block maintains identical dimensions and matrix properties, with only fracture geometries varying between blocks. The mesh refinement for both fractures and the matrix is adapted based on the local fracture density and complexity to ensure numerical accuracy.

Then, numerical simulations are conducted for each fractured grid block with pressure gradients applied along both x- and y-axes. The resulting flux distributions and pressure fields provide the necessary information for upscaling calculations. For grid blocks without fractures, the matrix permeability is directly assigned.

Subsequently, the MFU is applied to calculate the equivalent permeability for each grid block by using the previous simulation results. The MFU uses multiple boundary expressions to calculate flux. When the linear boundary conditions are applied along the x-axis, the flow rates along the x- and y-axes, $q_x$ and $q_y$, are calculated on multiple boundaries:

$$q_\mathrm{x} = \int_0^{l_y} v_\mathrm{r} \cdot n_\mathrm{x} \mathrm{d}y + \int_0^{l_\mathrm{x}} v_\mathrm{u} \cdot n_\mathrm{x} dx + \int_0^{l_\mathrm{x}} v_\mathrm{l} \cdot n_\mathrm{x} dx \ ,$$
$$q_\mathrm{y} = \int_0^{l_\mathrm{x}} v_\mathrm{u} \cdot n_\mathrm{y} dx + \int_0^{l_\mathrm{x}} v_\mathrm{l} \cdot n_\mathrm{y} dx + \int_0^{l_y} v_\mathrm{r} \cdot n_\mathrm{y} dy \ ,$$

(3)

where $l_x$ and $l_y$ are dimensions of the Cartesian grid in the x- and y-directions, $n_x$ and $n_y$ are unit vectors along the x- and y-axes, $v_\mathrm{r}$, $v_\mathrm{u}$, and $v_\mathrm{l}$ denote the Darcy velocities on the right, upper, and lower boundaries.

Lastly, the coarse-scale equivalent permeability can be computed inversely based on the Darcy's law using the flux in the fine-scale discrete fracture model. The details of the MFU can be found in Chen et al. (2015). The resulting permeability tensor components ($k_{xx}$, $k_{xy}$, $k_{yx}$, $k_{yy}$) characterize the anisotropic flow behavior of each grid block.

## 2.3 Algorithmic Framework and Implementation

The upscaling workflow is designed as a modular framework consisting of five interconnected computational stages, each addressing specific mathematical and numerical challenges in fractured porous rocks characterization. Table 2 summarizes the computational stages, their mathematical formulations, and key parameters.

**Table 2. Computational stages and mathematical framework of the UpsFrac workflow**

| Computational Stage | Mathematical Framework | Key Parameters | Primary Functions |
|---|---|---|---|
| DFM Generation | • Power-law distribution (Eq. 1) <br> • Aperture-length correlation (Eq. 2) | • num_real: number of DFM realizations <br> • n: number of fractures <br> • α: power-law exponent <br> • $\gamma$, D: aperture coefficients | Generate stochastic fracture realizations |
| Fracture-Coarse Grid Partitioning | • Geometric intersection algorithm <br> • Boundary clipping | • dx, dy: coarse grid dimensions <br> • tfrac: total fractures | Assign fractures to Cartesian blocks with boundary treatment |
| Simulation Setup | • TPFA/MPFA discretization <br> • Cubic law | • Mesh refinement level <br> • $k_\mathrm{m}$: matrix permeability | Create DFM for flow equations |
| Flow Simulation | • Darcy's law <br> • Orthogonal pressure gradients | •Boundary pressures <br> •Solver type | Compute directional flux fields |
| Upscaling Calculation | • multiple boundary approach (Eq. 3) <br> • Tensor assembly | • Flux decomposition angle <br> • Physical constraint checks | Calculate equivalent permeability tensor |

The framework incorporates key algorithmic features to ensure robust upscaling: (i) stochastic fracture generation with power-law length distributions and aperture-length correlations; (ii) geometric processing with boundary intersection detection and corner point adjustment to prevent mesh singularities; (iii) automatic mixed-dimension mesh generation integrating fracture and matrix elements; and (iv) batch processing of dual orthogonal flow problems. The upscaling employs directional flux decomposition through cosine transformations and validates permeability tensors to correct non-physical values from numerical errors. This modular architecture enables users to customize individual components—such as substituting alternative fracture generation models or numerical schemes—while maintaining framework compatibility, facilitating adaptation to diverse geological scenarios. The complete algorithmic pseudocode and implementation details are provided in Appendix A.1, with computational modules and data structures documented in Appendix A.2 for reproducibility.

## 2.4 Equivalent permeability visualization

The upscaled equivalent permeability tensors are analyzed and visualized to characterize flow behavior in fractured porous rocks. The resulting equivalent permeability is a full tensor form and is not inherently symmetric for fractured rocks (Chen et al., 2016; Zijl and Stam, 1992). When a symmetric tensor is required, a symmetric permeability tensor can be obtained by averaging the off-diagonal components. Using the upscaling results, the distribution of permeability tensor components ($k_{xx}$, $k_{xy}$, $k_{yy}$) can be visualized to characterize spatial heterogeneity and anisotropy. In addition, the equivalent permeability tensor of each grid block can also be plotted as an ellipse to visualize the directional permeability. Furthermore, to analyze the statistical properties of the equivalent permeability, histogram analysis of the permeability components is performed, which is important for building stochastic fields of equivalent permeability (Fiori et al., 2015) and supporting uncertainty quantification in reservoir modelling.

## 3 Validation

### 3.1 Fracture with power law length distribution

For modelling the fracture length with power law distribution, the cut-off power law is applied to create fracture length. For fractures with length l ranging from $l_{min}$ to $l_{max}$, the power law distribution can be expressed as in equation (1). The power law distribution for fracture length is created using the rndm_powerlaw function. The values of $l_{min}$, $l_{max}$, power-law exponent α and fracture number N should be determined. For instance, when $l_{min}$=10 m, $l_{max}$=1000 m, α=2.5, and N=500, the histogram of stochastically generated fracture length is plotted in Fig. 4, showing a typical power-law decay pattern with most fractures concentrated at smaller lengths. When fitting the stochastically generated data with the maximum likelihood estimation method, the fitted α is 2.52, close to the input alpha 2.5. The Kolmogorov-Smirnov statistic is 0.032, indicating that the software UpsFrac generates the data set and agrees with the fitted power-law distribution.

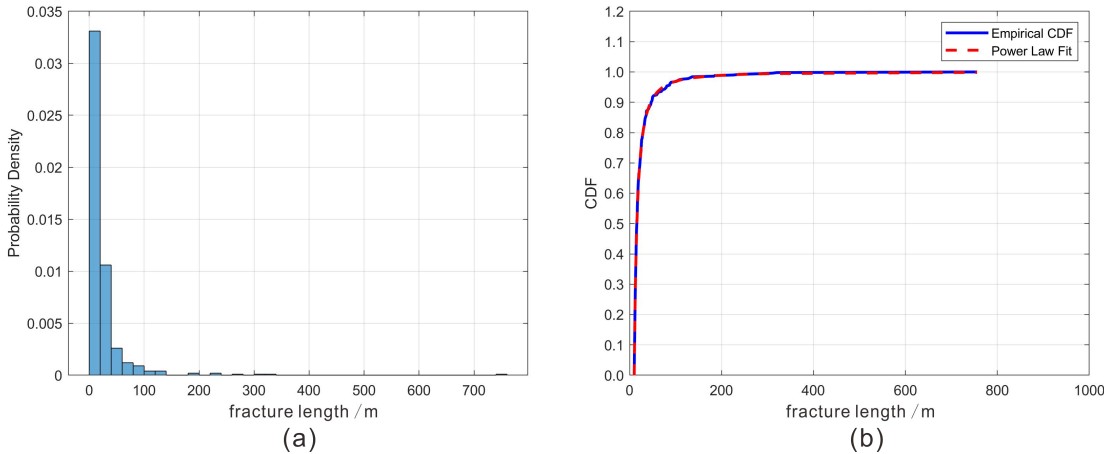

**Figure 4: (a) Histogram of power law distribution, (b) Cumulative Distribution Function (CDF) of created data and fitted power law.**

## 3.2 Equivalent fracture permeability with varied aperture

To validate the accuracy of UpsFrac's upscaling methodology, we compare the calculated equivalent permeability against analytical solutions for a single fracture embedded in a low-permeability matrix. The validation setup consists of a 2 m × 2 m domain containing a 2 m horizontal fracture at y = 1 m spanning the entire domain width (Fig. 5a). The matrix permeability $k_m = 9.87 \times 10^{-16}$ m$^2$, and the fracture permeability $k_f$ varies based on aperture according to the cubic law. Linear pressure boundary conditions are applied on all four boundaries: left boundary (x = 0) with P = 2 Pa, right boundary (x = 2) with P =

0 Pa, and top/bottom boundaries with P = 2 - x Pa, representing a unit gradient flow field.

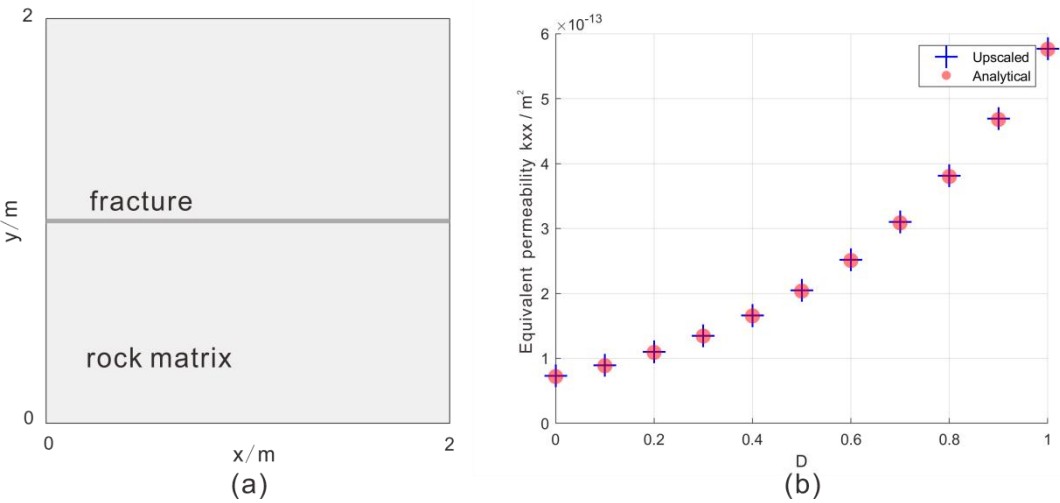

**Figure 5: Validation of upscaling methodology for a single horizontal fracture: (a) model geometry showing fracture and matrix configuration, and (b) comparison of upscaled and analytical equivalent permeability for varied aperture values.**

The analytical solution for equivalent permeability $k_{xx}$ under parallel flow assumption between fracture and matrix is derived from flow equivalence principle (Snow, 1969). For the single-fracture case, this can be simplified to:

$$k_{xx} = k_m \times (1 - w/\Delta y l_y) + k_f \times w/\Delta y l_y, \qquad (4)$$

where $k_f = w^2/12$ based on the cubic law, and $\Delta y$ is the domain height in the y-direction ($\Delta y = l_y$ for this single-block validation case).

We tested the upscaling accuracy by varying fracture aperture with correlation exponent D from 0 to 1 ($\gamma = 1.2 \times 10^{-4}$). The equivalent permeability calculated by UpsFrac ranged from $0.7 \times 10^{-13}$ m² to $5.8 \times 10^{-13}$ m², showing excellent agreement with the analytical solution (maximum relative error < 1%, Fig. 5b). The MFU enables accurate handling of fractures at arbitrary angles (Chen et al., 2015), critical for complex networks with non-aligned fractures. The modular framework allows easy incorporation of alternative aperture models, including rough fractures (see Section 5.1).

**3.3 Numerical accuracy and sensitivity analysis**

The numerical performance of the upscaling methodology is evaluated through three complementary analyses: grid convergence studies to establish discretization requirements, parametric investigation of fracture-matrix permeability contrasts, and orientation-dependent error assessment. These analyses collectively demonstrate the method's accuracy and robustness for practical applications.

**3.3.1 Grid convergence studies**

To assess the numerical accuracy and grid sensitivity of the upscaling methodology, systematic convergence studies were performed on a representative 2D fractured domain containing a single horizontal fracture. The computational domain measures 2 m × 2 m with the fracture positioned at y = 1 m, spanning the entire domain width. The fracture aperture was set to 120 µm with a permeability of $1.2 \times 10^{-9}$ m² based on the cubic law, while the surrounding rock matrix has a permeability

of $9.87 \times 10^{-16}$ m². This configuration represents a fracture-matrix permeability contrast of approximately six orders of magnitude, representing highly conductive fracture systems commonly encountered in crystalline rocks and tight formations. Six progressively refined grids were employed with spacings ranging from h = 1.0 m (2×2 grid) to h = 0.025 m (80×80 grid), enabling a comprehensive evaluation of grid-dependent effects on the computed equivalent permeability tensor (Fig. 6).

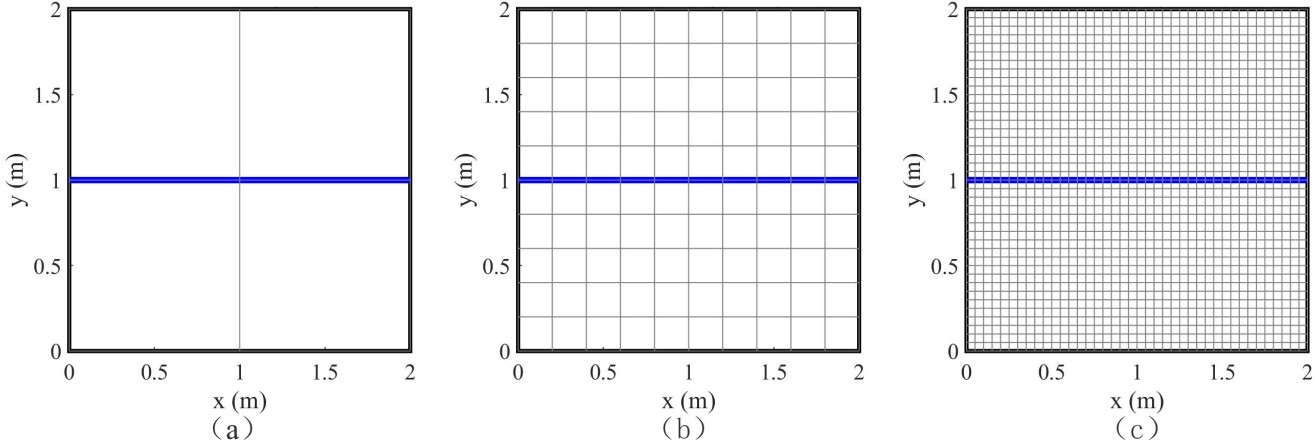

Figure 6: Visualization of selected grid refinement levels showing the progressive mesh resolution from (a) h = 1.0 m (2×2), (b) h = 0.2 m (10×10), to (c) h = 0.05 m (40×40) with the horizontal fracture indicated in blue.

The upscaling procedure demonstrates excellent convergence behavior with systematic grid refinement (Fig. 7a). The computed equivalent permeability values range from $7.2500 \times 10^{-14}$ m² for the coarsest grid to $7.2975 \times 10^{-14}$ m² for the finest resolution, converging monotonically. The relative errors reported here are computed against the finest grid solution (h = 0.025 m) as the reference to assess pure grid convergence behavior. Even the coarsest 2×2 grid achieves a relative error of only 0.65%, while grids with h ≤ 0.2 m maintain errors below 0.13%. The log-log error analysis (Fig. 7b) confirms rapid convergence with decreasing grid spacing. Notably, the h = 0.1 m grid (20×20) provides accuracy within 0.052% while using only 6% of the computational cells compared to the finest 80×80 grid. These results establish that grid spacings of 0.1-0.2 m offer optimal balance between computational efficiency and numerical accuracy for practical applications.

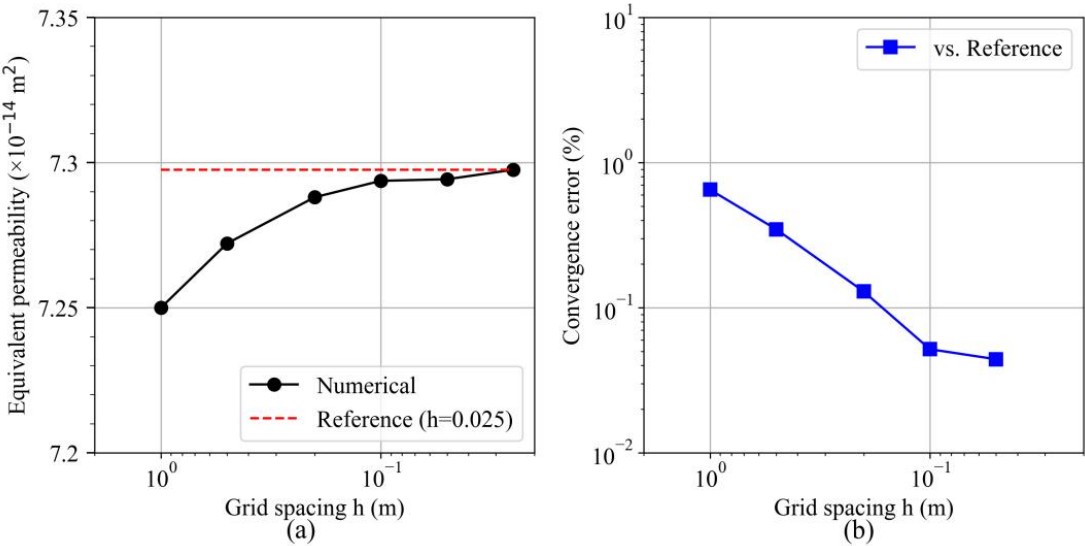

Figure 7: Grid convergence of equivalent permeability showing (a) permeability values; (b) relative error.

### 3.3.2 Fracture-Matrix permeability contrast

The influence of fracture-matrix permeability contrast on upscaling accuracy was investigated using a 2 m × 2 m domain with a single horizontal fracture. The matrix permeability, $k_m$, was fixed at $9.87 \times 10^{-16}$ m², while fracture aperture varied from 1.09 to 108.83 μm, yielding fracture permeabilities, $k_f$, from $9.9 \times 10^{-14}$ to $9.87 \times 10^{-10}$ m² based on the cubic law. This creates permeability contrasts spanning four orders of magnitude ($k_f/k_m \approx 10^2$ to $10^6$), encompassing conditions from tight formations to highly conductive systems. The equivalent permeability tensor was computed using UpsFrac and compared against analytical solutions from parallel flow theory.

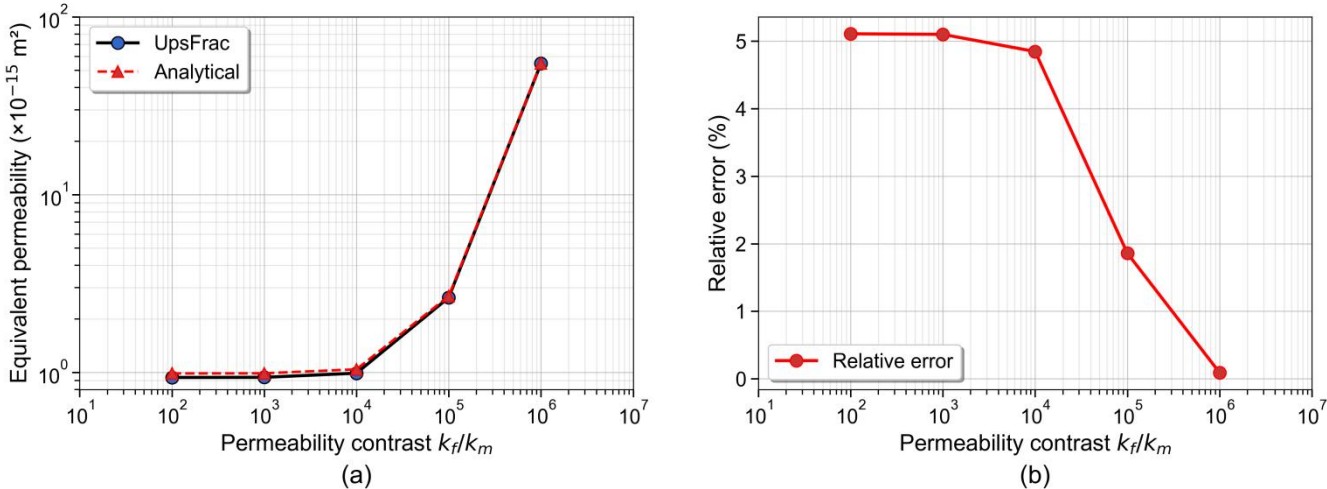

(a)             (b)

**Figure 8: Comparison of upscaled equivalent permeability between UpsFrac numerical solutions and analytical reference values for different fracture-matrix permeability contrasts (a) comparison of equivalent permeability; (b) relative error analysis.**

Figure 8a demonstrates excellent accuracy across the entire contrast spectrum. The upscaled permeability values closely track analytical solutions, capturing the transition from matrix-dominated ($k_f/k_m < 10^3$) to fracture-dominated ($k_f/k_m > 10^5$) flow regimes. At low contrasts, equivalent permeability approximates the matrix value; at high contrasts, it reaches several orders of magnitude above the matrix. The numerical solutions consistently capture this nonlinear scaling with remarkable fidelity.

Figure 8b reveals consistently low relative errors across all contrasts, with a mean error of 3.40%. The maximum error (5.11%) occurs at low contrasts where fracture contribution is minimal, decreasing to <0.1% for high-contrast systems. This trend confirms UpsFrac's suitability for highly fractured systems requiring accurate preferential flow representation. The robust performance across diverse contrasts validates the upscaling formulation for applications ranging from tight shales to karstified carbonates.

### 3.3.3 Orientation-dependent error

To evaluate the sensitivity of the upscaling methodology to fracture orientation relative to the computational grid, a comprehensive analysis was conducted using a single fracture rotated at angles of 0°, 30°, 45°, 60°, and 90° within a 2 m × 2 m domain (Fig. 9). The fracture maintains a constant aperture of 120 μm, embedded in a matrix with permeability of $9.87 \times 10^{-16}$ m². For each orientation, the full permeability tensor was computed using UpsFrac and compared against analytical solutions obtained through coordinate transformation of the reference configuration. For a fracture with azimuth $\theta$, the equivalent permeability tensor can be written as:

$$k(\theta) = \begin{bmatrix} k_{xx} & k_{xy} \\ k_{yx} & k_{yy} \end{bmatrix} = \begin{bmatrix} k_x^* \cos^2 \theta + k_y^* \sin^2 \theta & (k_x^* - k_y^*) \cos\theta \cdot \sin\theta \\ (k_x^* - k_y^*) \cos\theta \cdot \sin\theta & k_x^* \sin^2 \theta + k_y^* \cos^2 \theta \end{bmatrix}, \tag{5}$$

where $k_x^*$ and $k_y^*$ are the equivalent permeability tensor components in the x- and y-directions for a horizontal fracture ($\theta = 0°$), respectively. Here, $k_x^*$ can be calculated from Eq. 4, and $k_y^* \approx k_m$ since a single fracture aligned with the x-axis has negligible impact on the y-direction permeability.

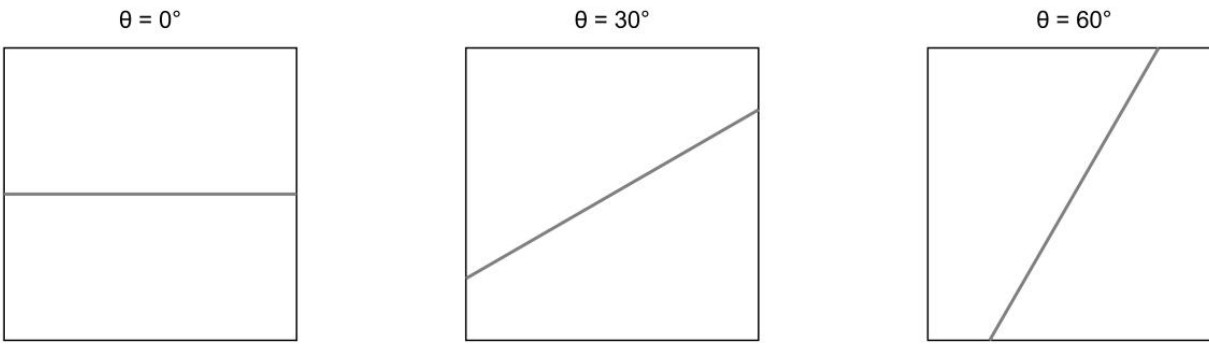

**Figure 9: Representative fracture orientations at (a) 0°, (b) 30°, and (c) 60°.**

Figure 10a demonstrates the evolution of permeability tensor components with fracture orientation. The diagonal components $k_{xx}$ and $k_{yy}$ exhibit sinusoidal variation between fracture-dominated ($7.3 \times 10^{-14}$ m²) and matrix-dominated ($9.87 \times 10^{-16}$ m²) directions, while $k_{xy}$ peaks at 45° where fracture contribution is equally distributed between the x- and y-directions. The UpsFrac solutions closely match analytical curves for this single fracture configuration, achieving excellent agreement with mean relative errors of 1.10%, 3.15%, and 0.07% for $k_{xx}$, $k_{yy}$, and $k_{xy}$, respectively. The accurate calculation of off-diagonal components is particularly important when sparse fractures are not aligned with the coordinate axes.

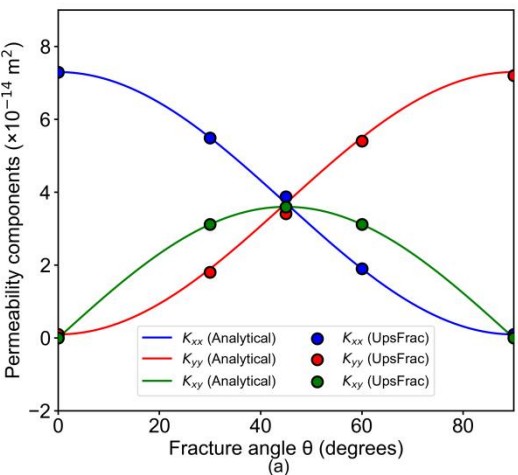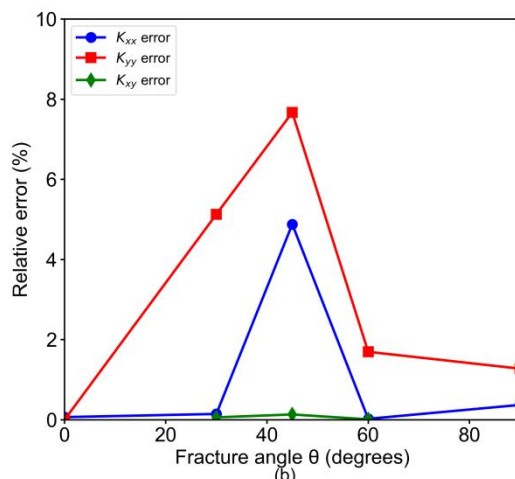

**Figure 10: Orientation-dependent validation of upscaled permeability: (a) permeability tensor components as functions of fracture orientation angle; (b) relative errors between UpsFrac numerical solutions and analytical reference values.**

Figure 10b reveals the maximum error (7.67%) occurs at 45° orientation. At 45° orientation, fracture endpoints naturally terminate near domain corner nodes, creating ambiguity in boundary condition assignment. To maintain numerical stability and avoid singular matrix configurations, slight geometric adjustments are required to displace fracture endpoints away from exact corner positions. Despite this localized issue, overall accuracy remains excellent with errors below 5% for most orientations. These results validate that UpsFrac reliably handles arbitrarily oriented fractures without grid alignment, essential for modelling complex natural fracture networks.

## 4 Software application

### 4.1 Discrete fracture modelling and upscaling application

The modelling and upscaling procedure is applied to a real-scale problem by using UpsFrac. The model domain is 1000 m × 1000 m. The domain contains both large-scale fractures (faults) and small-scale fractures, referring to the measured data in the field (Massart et al., 2010). The fracture network contains both deterministic fractures and stochastically generated fractures. The deterministic fractures include three fractures with an aperture of 0.001 m. The stochastic fracture length follows a power law distribution, with $l_{min}$=10 m, $l_{max}$=1000 m, N=200, and $\alpha$=2.5. Fracture orientation follows Fisher distribution with $\kappa$ =0, meaning the fracture orientation is randomly distributed. Fracture location follows uniform distribution. In this study, the aperture is correlated with fracture length with $\gamma$ of $2.3\times 10^{-5}$ and D of 0.5 according to the field data (Schultz and Soliva, 2012). The above geometric data are input parameters for the fracture modelling module. The rock matrix permeability is homogeneous and isotropic with $9.87\times10^{-16}$ m$^2$. One realization of the discrete fracture model is shown in Fig. 11.

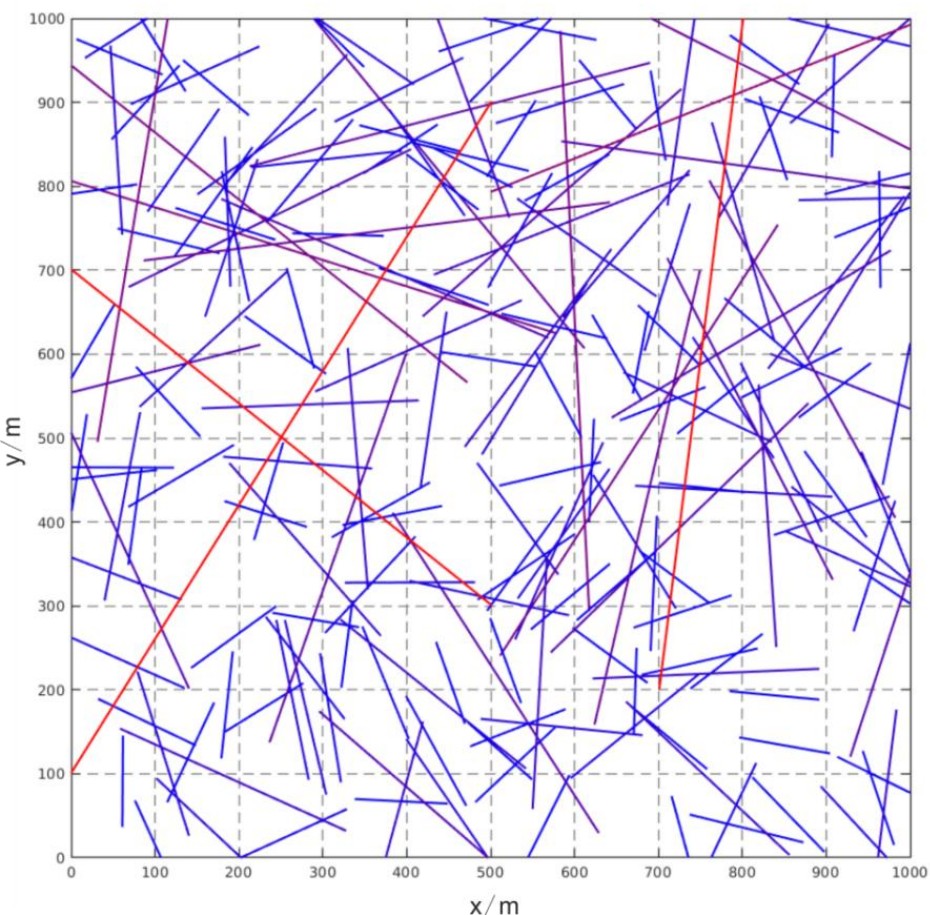

**Figure 11: Discrete fracture model generated by UpsFrac. Blue denotes a low aperture, red denotes a high aperture, and the dashed line denotes the grid blocks on a coarse scale.**

The grid block size for upscaling is 100 m×100 m, which is the input parameter for the grid subdivision module. Each grid contains 3-9 fractures clipped by the grid boundaries. For the numerical simulation of each grid, the unstructured mesh size 330 for the rock matrix is 5 m, and for fracture, it is 3.3 m. After simulating fluid flow in all grid blocks, the required information for upscaling is calculated. Through the upscaling calculation module, the equivalent permeability for each grid block is calculated.

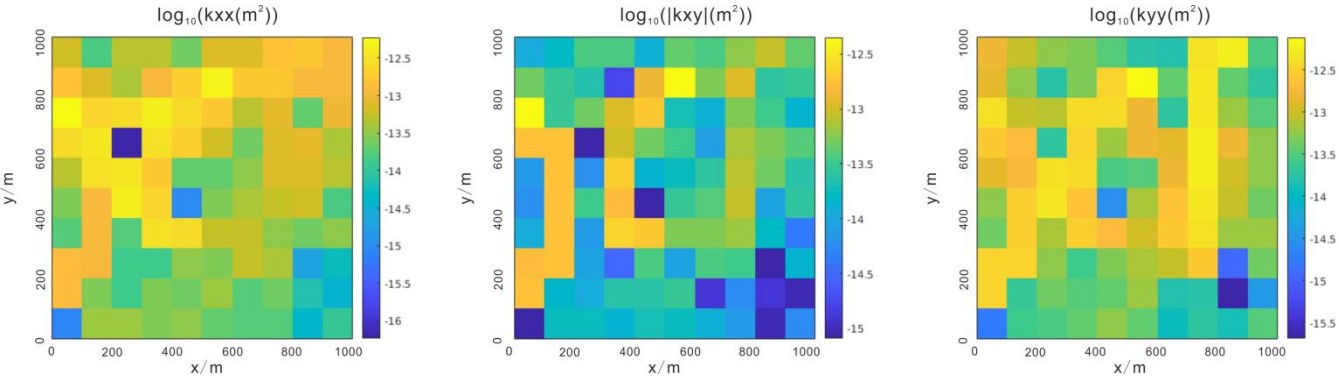

**Figure 12: Equivalent permeability distribution of grid blocks on a coarse scale.**

The resulting equivalent permeability distribution is visualized in Fig. 12. The equivalent permeability distribution shows that it is highly correlated with the fracture network geometries of fractures (Fig. 12). The anisotropy in equivalent permeability components is controlled by the fracture orientation distribution.

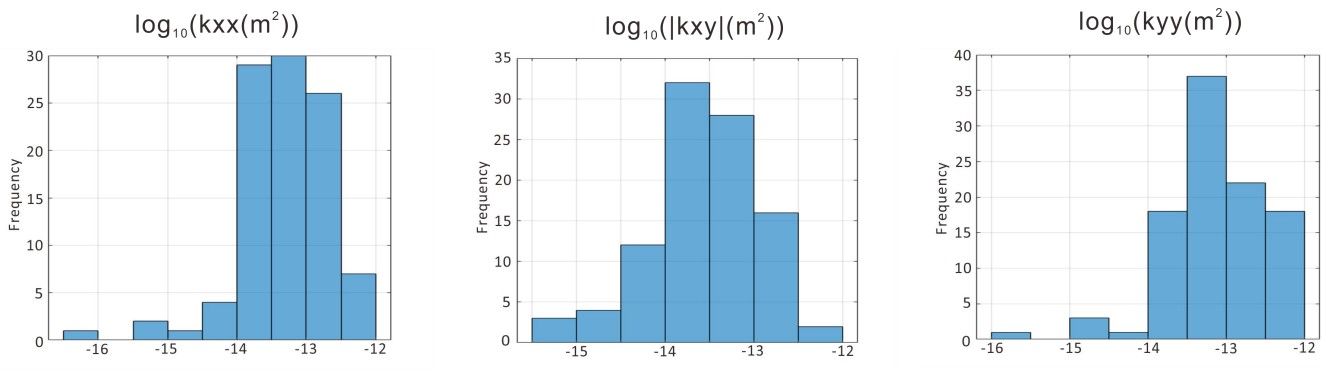

**Figure 13: Histograms of upscaled equivalent permeability components.**

The histograms of different components of equivalent permeability are plotted in Fig. 13. They show that all components $k_{xx}$, $k_{xy}$, and $k_{yy}$ follow a log-normal distribution. For $k_{xy}$, it can be positive and negative; we use the absolute value here for plotting. For $k_{xx}$ and $k_{xy}$, the shapes of the histograms are similar. The results indicate that even though the spatial distribution is different, the statistical properties of $k_{xx}$ and $k_{xy}$ are similar. This is mainly due to the random distribution of fracture orientations, i.e., without a preferable orientation. For $k_{xy}$, it is slightly lower than $k_{xx}$ and $k_{xy}$; this is consistent with the positive-definite nature of the permeability tensor (Lang et al., 2014).

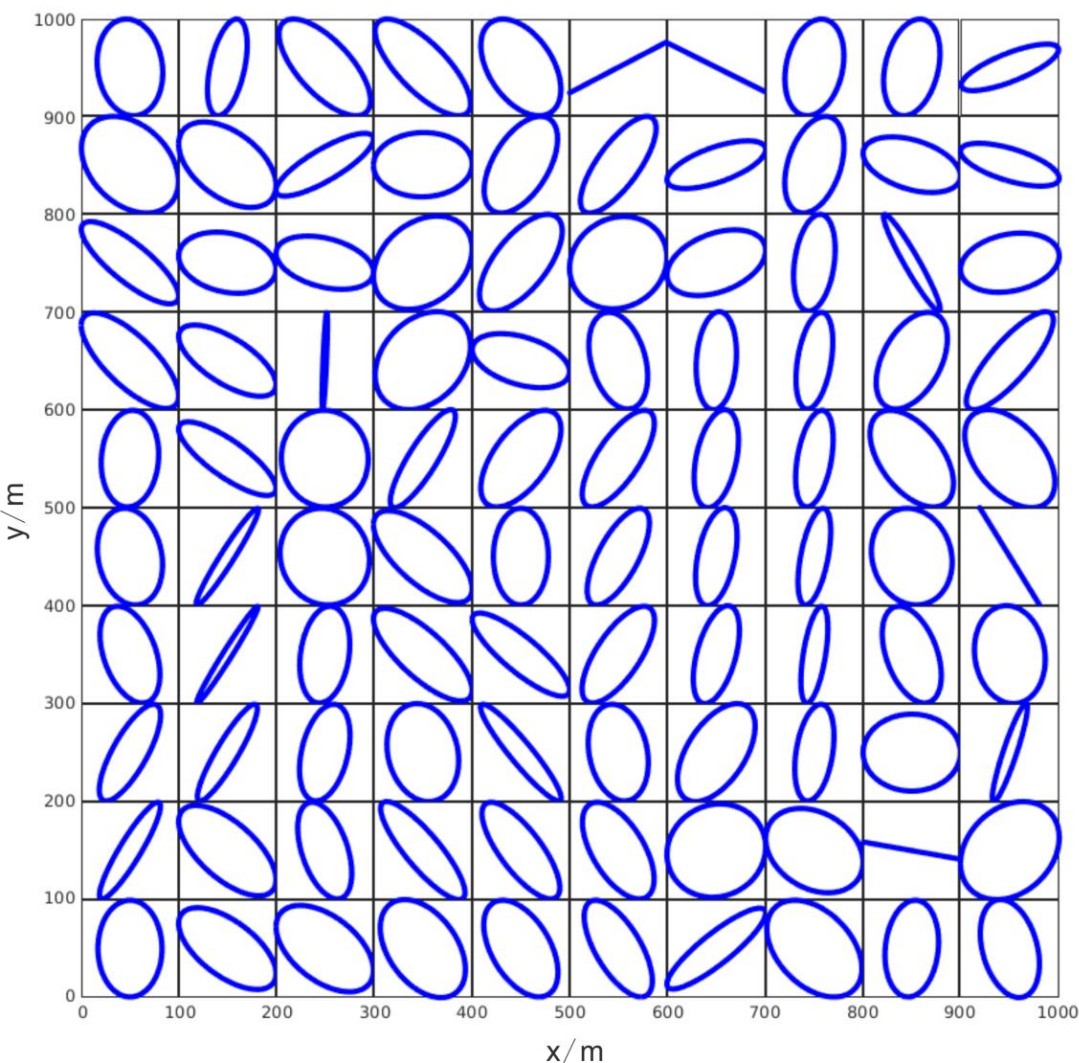

**Figure 14: Ellipse of equivalent permeability tensor for the model. Ellipses are normalized within each subplot to enhance visualization of tensor orientation rather than absolute permeability magnitude.**

The permeability tensor can be visualized using the ellipses. The permeability ellipses for all grid blocks are plotted in Fig. 14. They show that the long axes of an ellipse follow the fracture orientations (Fig. 11), especially for the grid with a few large fractures having high permeability (e.g., the grid block near the lower left corner of the model). The consistency between the upscaled permeability tensor ellipses and the fracture network geometries indicate that the upscaling code UpsFrac yields reasonable and accurate results. These results can be used to characterize the permeability of the fractured porous rocks and serve as input to equivalent fracture models.

## 4.2 Comparison between upscaled EFM and DFM for fractured geothermal systems

A two-dimensional fractured geothermal system model was developed to evaluate the computational performance and accuracy of the EFM upscaling approach (Fig. 15). The model simulates a 500 m × 500 m geothermal reservoir with a doublet injection-production well configuration (150 m spacing), where cold water at 40 °C is continuously injected at the injection well and produced at the production well, both at 1 L/s. The domain is initialized at 80°C and 20 MPa, with no-flow and adiabatic boundary conditions representing sealed cap and base rocks. The rock matrix has typical geothermal reservoir properties: permeability of $9.87 \times 10^{-16}$ m², 1% porosity, density of 2600 kg/m³, specific heat of 950 J/(kg·K), and thermal conductivity of 3.0 W/(m·K). The fractures feature a 500 μm mean aperture with 100% porosity. Fluid properties include density of 1000 kg/m³, viscosity of 0.001 Pa·s, specific heat of 4200 J/(kg·K), and thermal conductivity of 0.65 W/(m·K), representing typical values for fractured geothermal systems.

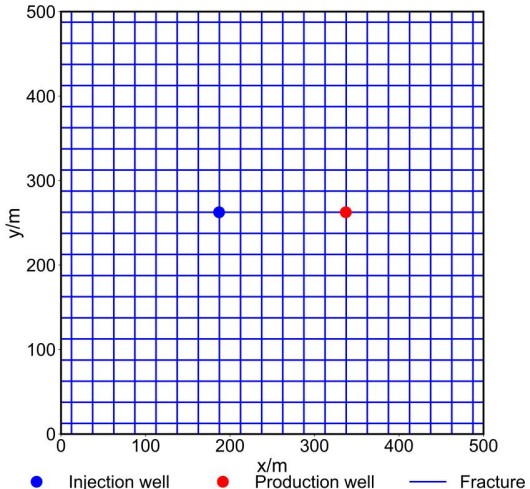

**Figure 15: Conceptual model of the field-scale geothermal doublet system with 500 m × 500 m domain and 150 m well spacing.**

The DFM and EFM models were constructed using different meshing strategies to enable direct performance comparison (Fig. 16). For the DFM model, GMSH was employed to generate an unstructured mesh with 13,414 nodes and 30,006 elements, comprising 3,820 line elements for the fracture network and 26,186 triangular elements for the rock matrix. In contrast, the EFM model utilized a structured 20×20 grid with uniform cell dimensions of 25 m × 25 m, resulting in 400 grid cells. The upscaling procedure using UpsFrac yielded homogeneous equivalent permeabilities of $k_{xx} \approx 4.116 \times 10^{-13}$ m² and $k_{yy} \approx 4.100 \times 10^{-13}$ m², with zero off-diagonal terms ($k_{xy} = k_{yx} = 0$), indicating isotropic permeability characteristics and alignment of the principal permeability directions with the coordinate axes.

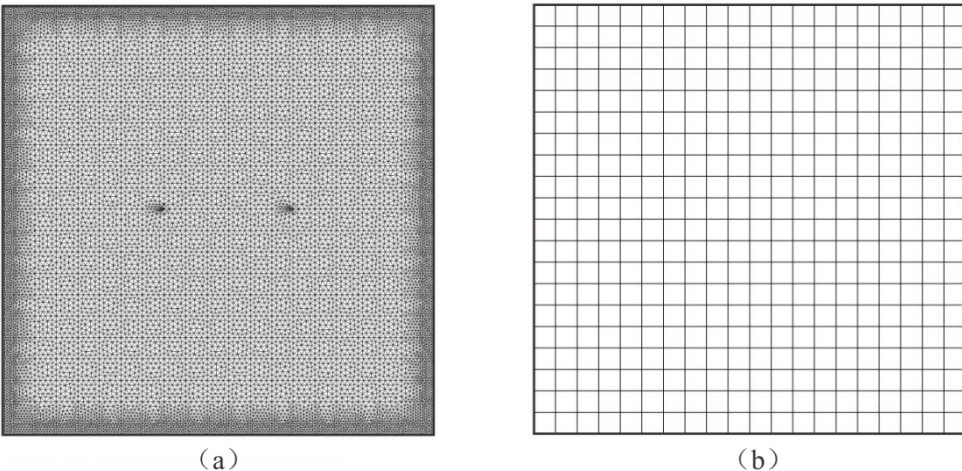

(a)                                      (b)

**Figure 16. Mesh discretization for (a) DFM model with 13,414 nodes and (b) EFM 20×20 structured grid.**

Both EFM and DFM were employed to solve the coupled flow and heat transfer processes using SHEMAT-Suite and OpenGeoSys, respectively. The validation results demonstrate excellent agreement between the upscaled EFM model and the high-fidelity DFM benchmark (Fig. 17). Temperature breakthrough curves at the production well over 30 years show mean relative error of approximately 0.2%, achieved with only 400 coarse grid cells compared to the 13,414-node DFM discretization, confirming the upscaling approach's high accuracy and robustness.

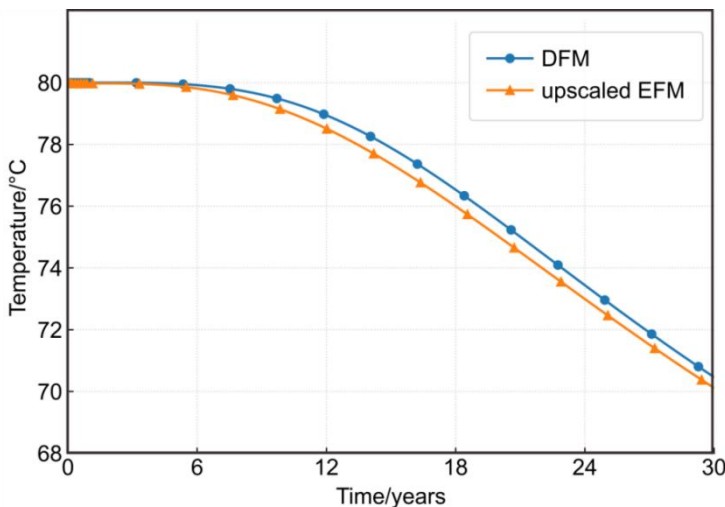

**Figure 17: Temperature breakthrough curves comparing DFM reference and EFM upscaled models over 30 years.**

The computational performance analysis reveals substantial efficiency gains of the EFM approach (Fig. 18). The EFM upscaling process consists of four stages: Fracture-Coarse Grid Partitioning (0.186 s, 0.13 MB), Simulation Setup (28.659 s, 3.02 MB), Flow Simulation (71.245 s, 13.03 MB), and Upscaling Calculation (49.650 s, 0.67 MB), totaling 149.74 s for the upscaling process. The subsequent EFM simulation requires 540 s and 7.48 MB. In contrast, DFM using OpenGeoSys requires meshing (21.45 s, 139.41 MB) and simulation (2081 s, 75.30 MB). The complete EFM workflow required only 689.74 s versus 2102.45 s for DFM—a 67% reduction. Total memory usage decreased from 214.71 MB to 24.33 MB—an 89% reduction. These benchmarks were conducted on an Intel Core i5-9300H system with 8 GB RAM. The results confirm that the EFM upscaling methodology achieves both computational efficiency and numerical accuracy (~0.2% relative error), making it particularly suitable for large-scale fractured reservoir simulations. However, accuracy may decrease for complex fracture networks with highly irregular geometries or dense intersection patterns, where validation against fine-scale DFM solutions or field data becomes essential.

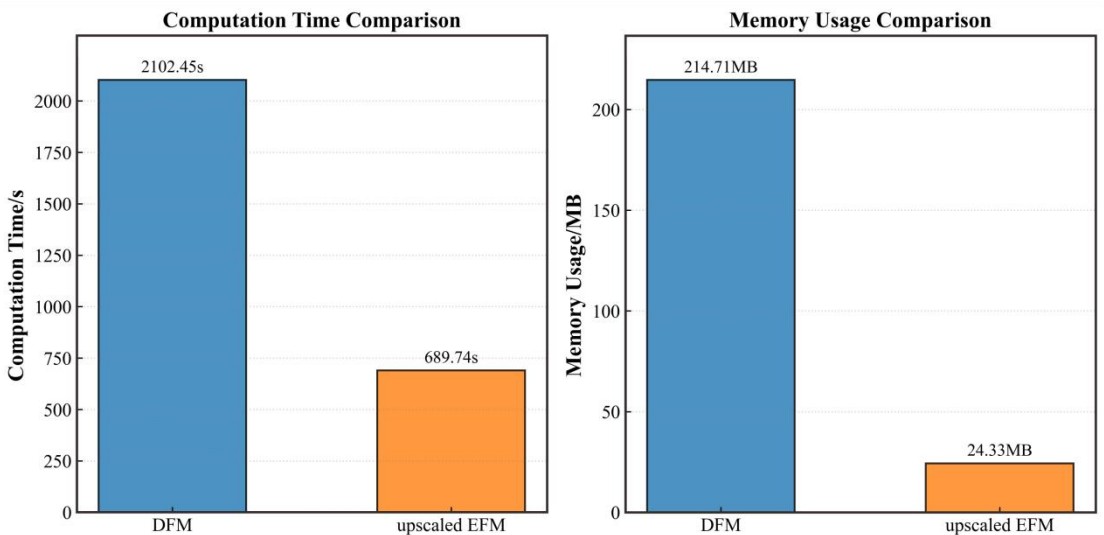

**Figure 18: Computational performance comparison showing runtime and memory usage for DFM and EFM simulations.**

### 4.3 Stochastic Ensemble Analysis and Uncertainty Quantification of Upscaled Permeability

To quantify uncertainty in upscaled properties arising from stochastic DFN realizations, we performed Monte Carlo analysis with 100 independent realizations using the same geometric parameters, matrix properties, and grid configuration as described in Section 4.1. The ensemble analysis reveals distinct distribution characteristics for the upscaled permeability tensor components. The diagonal components ($k_{xx}$ and $k_{yy}$) exhibit approximately log-normal distributions as confirmed by

Q-Q plot analysis, while the off-diagonal component ($k_{xy}$) shows normal distribution characteristics (Fig. 19). The mean upscaled permeability values across all realizations and grid cells are $k_{xx} = 1.334 \times 10^{-13}$ m², $k_{yy} = 1.325 \times 10^{-13}$ m², and $k_{xy} = 6.685 \times 10^{-16}$ m². The coefficients of variation reveal moderate variability for diagonal components ($k_{xx}$: CV=0.889, $k_{yy}$: CV=0.910) and significantly higher uncertainty for the off-diagonal component ($k_{xy}$: CV=106.288), primarily due to its very small mean value ($6.685 \times 10^{-16}$ m²) relative to its standard deviation ($7.105 \times 10^{-14}$ m²), resulting in an amplified coefficient of variation.

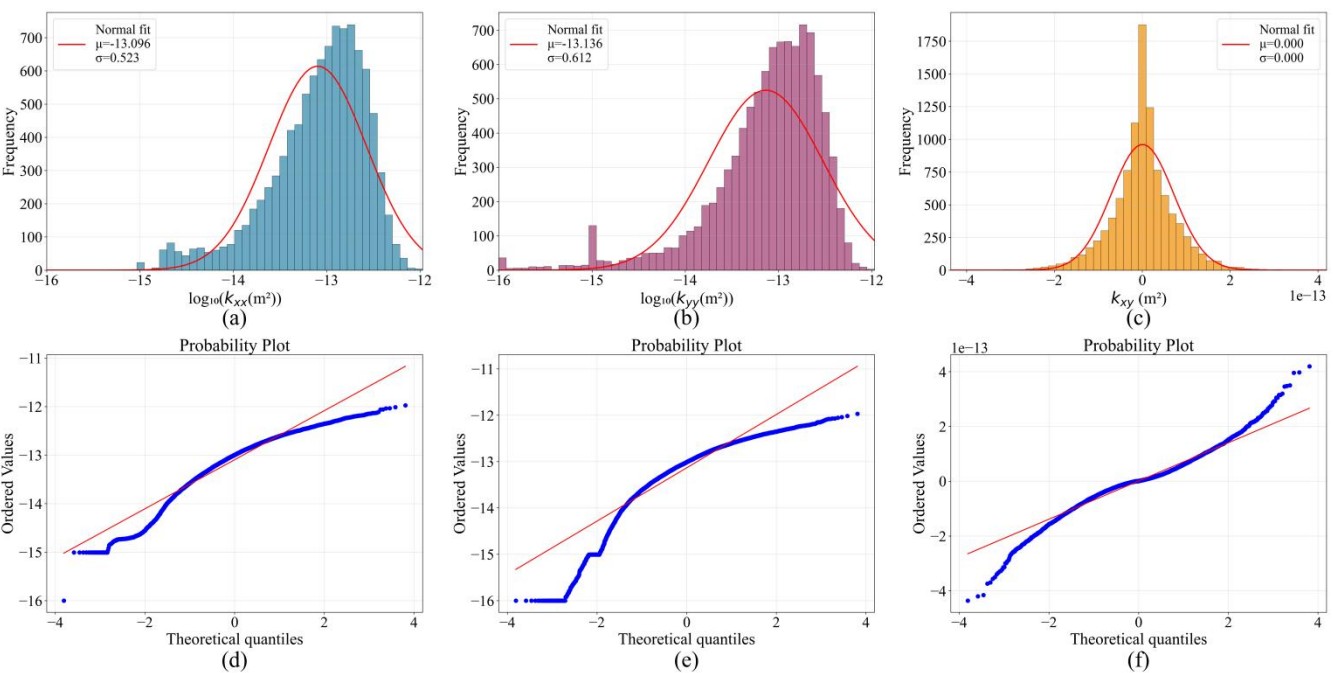

**Figure 19: Distribution characteristics of upscaled permeability tensor components: (a,c,e) histograms and (b,d,f) Q-Q plots for Kxx, Kyy, and Kxy respectively.**

Convergence analysis demonstrates robust statistical behavior of the ensemble approach. The mean values of diagonal permeability, $k_{xx}$ and $k_{yy}$, components stabilize after approximately 50-60 realizations, achieving relative errors below 5% compared to the full 100-realization ensemble, all converging to $1.33 \times 10^{-13}$ (Fig. 20). The off-diagonal component $k_{xy}$ converges to near-zero values. Due to the lack of pronounced directional preference in the fracture orientation distribution, and additionally, the relatively small magnitude of $k_{xy}$ values leads to slightly larger convergence relative errors compared to $k_{xx}$ and $k_{yy}$. This convergence pattern indicates that 50 realizations provide sufficient statistical reliability for practical uncertainty quantification, offering guidance for balancing computational cost with accuracy requirements. The rapid convergence of ensemble statistics validates the Monte Carlo approach for capturing the stochastic variability inherent in fractured porous rocks.

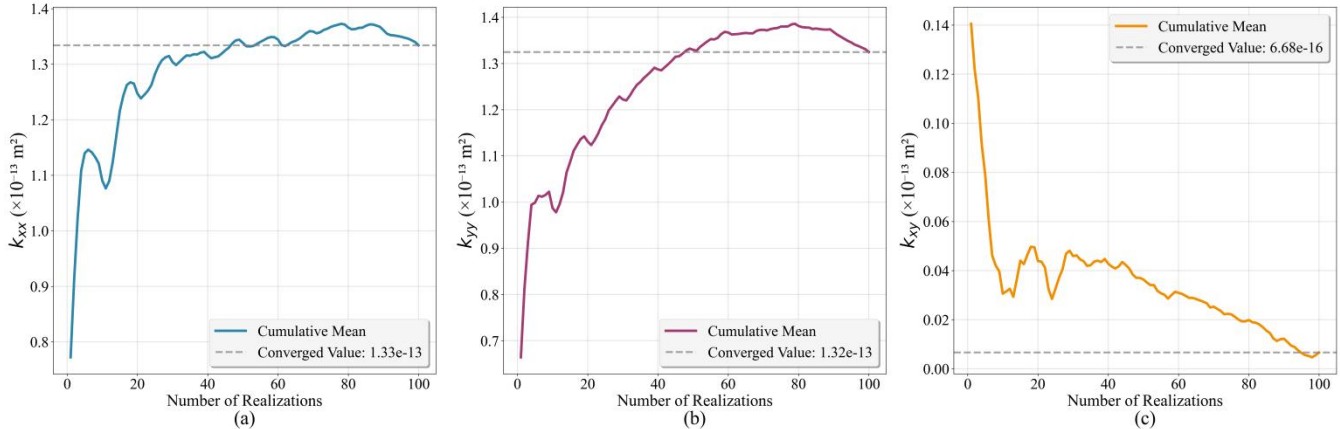

Figure 20: Convergence of ensemble mean values for upscaled permeability components: (a) kxx, (b) kyy, and (c) kxy.

The computational performance analysis reveals that the complete workflow—from fracture network generation to permeability upscaling—requires approximately 108 minutes total computational time for 100 realizations (1.08 minutes per realization) with peak memory usage of 8.2 GB. These benchmarks were conducted on a workstation equipped with an Intel Xeon W-2102 CPU @ 2.90 GHz and 16 GB RAM, running 64-bit Windows 7. The workflow consists of five stages: DFM Generation (0.19 s/realization), Fracture-Coarse Grid Partitioning (0.93 s/realization), Simulation Setup (1.43 s/realization), Flow Simulation (40.4 s/realization), and Upscaling Calculation (17.9 s/realization). Flow Simulation dominates the computational cost, accounting for 65% of total runtime, which is expected given the need to solve pressure equations for accurate effective permeability calculations. This ensemble-based uncertainty quantification demonstrates UpsFrac's capability to efficiently perform Monte Carlo workflows for stochastic DFN analysis, transforming complex discrete fracture systems into stochastic continuum representations while preserving uncertainty information critical for risk assessment in subsurface applications.

## 5 Limitations and future developments

### 5.1 Impact of constant aperture assumption on fracture roughness modelling

The constant aperture assumption, while computationally efficient and widely adopted in discrete fracture modelling, represents a significant simplification of natural fracture geometries. Real fractures exhibit complex aperture distributions arising from surface roughness, stress-induced deformation, and mineral precipitation or dissolution processes (Li et al., 2022; Xue et al., 2022). To quantify the impact of this simplification, we conducted a systematic analysis comparing upscaled permeabilities computed using constant versus variable aperture distributions. The test case employs a single horizontal fracture configuration identical to the validation case (Fig. 9, $\theta = 0°$) with a 2 m × 2 m domain and mean aperture $w_0 = 1.2 \times 10^{-4}$ m. Fig. 21a demonstrates the aperture variability along a 2 m fracture for different roughness levels

characterized by coefficients of variation (CV) ranging from 0.1 to 0.5, representing mild to severe aperture heterogeneity. The corresponding statistical distributions of these aperture profiles are shown in Fig. 21b through box plots, revealing increasing spread and skewness with higher CV values.

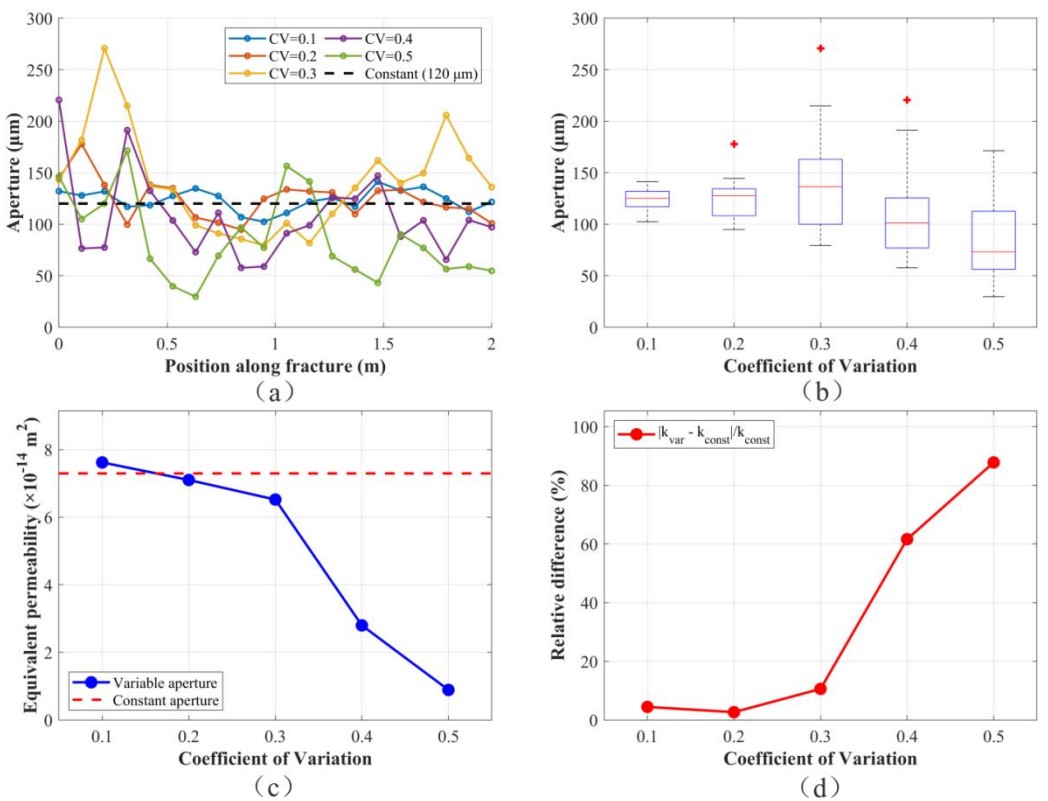

Figure 21: Effects of aperture variability: (a) aperture profiles along a 2 m fracture (CV = 0.1–0.5, mean aperture $w_0$ = 1.2×10-4 m); (b) aperture statistics (box: IQR; red line: median; whiskers: 1.5×IQR; crosses: outliers); (c) equivalent permeability vs CV (relative to constant aperture); (d) relative error of the constant-aperture assumption.

Our analysis exposes fundamental limitations of the constant-aperture assumption. Although it enables tractable simulations, accuracy degrades markedly for naturally rough fractures. As illustrated in Fig. 21c, the upscaled permeability in the x-direction ($k_{xx}$) decreases substantially with increasing roughness relative to the constant aperture case. Errors increase nonlinearly with roughness. Figure 21d quantifies this degradation, showing relative errors rising from negligible (<5%) when CV < 0.2 to severe (>60%) when CV > 0.4. This occurs because the arithmetic-mean aperture cannot capture flow channeling, and the cubic law disproportionately amplifies contributions from wider segments while constrictions strongly restrict flow.

These findings have direct implications for reservoir characterization and model selection. For preliminary assessments or systems dominated by smooth, fresh tensile fractures in crystalline rocks or well-sorted sedimentary formations (CV < 0.2), where Fig. 21d indicates errors remain below 5%, the constant-aperture model offers an efficient and sufficiently accurate approximation.In contrast, for highly rough fractures such as sheared zones and mineralized joints, explicitly representing aperture variability is essential for reliable predictions. UpsFrac's modular architecture readily accommodates this extension:

supporting rough fractures requires only augmenting fracture property storage and revising local flow calculations, while preserving the existing upscaling framework.

## 5.2 Three-dimensional model extension roadmap

While the current 2D implementation provides a robust foundation for method validation and pedagogical purposes, extending to 3D is essential for industrial applications. To address industrial needs, we outline a comprehensive roadmap for

extending UpsFrac to 3D. Specifically, we will: (i) generalize DFN generation from 2D line fractures to 3D planar polygonal or triangulated fracture surfaces with realistic size, orientation, aperture, and connectivity distributions; (ii) extend MFU to 3D with robust treatment of fracture intersections, complex fracture–matrix exchange, anisotropy, and boundary conditions; and (iii) mitigate the substantially higher computational cost through algorithmic accelerations, including adaptive/octree meshing, hierarchical fracture-surface meshing, algebraic multigrid preconditioning, domain decomposition with parallel

CPU/GPU solvers, and graph-based pruning of disconnected subnetworks. Building on our prior 3D upscaling experience (Chen et al., 2018) and established platforms such as ADFNE and MRST, we anticipate that the integration is technically feasible, though computational challenges primarily arise from the cubic scaling of 3D problems—where doubling the grid resolution increases computational cost eightfold. This provides a clear path toward evaluating industrial applicability in complex reservoir systems.

## 5.3 Accelerating flow-based upscaling

UpsFrac implements a theoretically rigorous flow-based upscaling approach. In practice, the upscaling step requires numerically solving flow in a discrete fracture model, where numerical stability must be considered for complex geometries. Achieving stable and accurate solutions may require adjusting parameters such as the mesh resolution in the fractures and the rock matrix. To prevent failed runs and facilitate correction, grid blocks that contain no fractures or that fail to converge are

485 logged for review; by reviewing these logs, users can refine global template inputs or adjust settings for specific blocks. In the future, data-driven surrogates trained on high-fidelity DFN simulations (e.g., Almajid and Abu-Al-Saud, 2022; Yan et al., 2024) may augment or replace direct solves to further improve robustness and efficiency.

## 5.4 Integration of alternative upscaling methods

The UpsFrac software applies MFU for calculating equivalent permeability, which is accurate as a flow-based method; other

fast analytical upscaling methods (Ebigbo et al., 2016; Oda, 1985), such as Effective Medium Theory (EMT), can also easily be implemented within the UpsFrac framework to upscale permeability in fractured porous rocks. The software can be further developed to broaden its applicability and improve user convenience. Key areas for improvement include the user interface, software stability, and compatibility with other reservoir simulation tools. This can be achieved through standardization of data formats and the development of API interfaces, which will facilitate broader adoption in industrial settings (Yan et al., 2024).

## 6 Conclusions

The paper describes a new free and open-source software UpsFrac for modelling and upscaling the permeability of two-dimensional fractured porous rocks with fractal characteristics. The software provides a methodology for creating multiple-scale fractures with varied apertures using both deterministic and stochastic approaches. The complex relations between fracture length, fracture density, and fracture aperture can be easily implemented in the framework of UpsFrac. The software applies the state-of-the-art upscaling method, MFU, which is a flow-based upscaling approach based on the TPFA or MPFA schemes and provides accurate and robust upscaling results.

UpsFrac addresses a critical gap in modelling of fractured porous rocks by providing an integrated open-source platform that seamlessly connects fractal fracture characterization with practical upscaling workflows. The software's key innovations include automated uncertainty quantification through Monte Carlo ensembles, direct DFM-to-EFM coupling without intermediate file conversions, and a modular architecture enabling algorithm substitution. Validation demonstrates exceptional accuracy across diverse geological conditions while achieving significant computational efficiency gains.

The software provides a flexible toolbox for modelling and analyzing fractured porous rocks with complex fracture geometries. The discrete fracture models can be upscaled to calculate equivalent permeability. The software is useful for investigating the heterogeneity and anisotropy of equivalent fracture permeability for fractured porous rocks. The upscaled permeability tensors can serve as input parameters in the equivalent fracture models. All the code and application scripts are available for download.

## Author contribution

TC: Conceptualization, Writing-original draft, Software, Investigation, Formal analysis, Funding acquisition. HS: Writing-review & editing, Methodology, Validation, Visualization. YZ: Writing-review & editing, Software, Data curation. FK: Writing-review & editing.

**Competing interests**

The authors declare that they have no competing interests.

**Acknowledgments**

We sincerely thank Mohammad Sabah and one anonymous reviewer for their valuable comments and constructive suggestions on the manuscript. Their careful reading and insightful feedback have substantially improved the quality of this paper. This work was funded by the National Natural Science Foundation of China (42002261). We would like to extend our

gratitude to the open source software developers related to ADFNE and MRST. We thank Anozie Ebigbo for valuable discussions and constructive suggestions on the manuscript.

**Code and data availability**

The UpsFrac code is available on GitHub (https://github.com/chentao9330/UpsFrac, last access: 30 September 2025), and version 1.0 is archived on Zenodo (Chen, 2025; https://doi.org/10.5281/zenodo.14674083). The UpsFrac code is published

under the GNU General Public License v3.0 (GPL-3.0 license). All input data in this study can be reproduced following the parameters and procedures described in this paper. The example input file used in this study can be found in the example folder of UpsFrac.

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

**Appendix A: Algorithm Implementation and Technical Details**

**A.1 Core Upscaling Algorithm**

---

```
     Algorithm: UpsFrac Equivalent Permeability Upscaling
Main input:
        - Domain parameters: L (domain size), nx, ny (grid numbers), d (grid block size)
        - DFN parameters: n (fracture number), l_min, l_max, α (power-law exponent)
        - Matrix permeability: k_matrix
        - Number of realizations: num_real
     Output:
        - Equivalent permeability tensor K_eq[i,j] for each grid block

     Main Procedure:
     1. DFN Generation (DiscreteFractureModeling.m):
        FOR each realization r = 1 to num_real:
           Generate n fractures with:
              - Length l[i] ~ power_law(l_min, l_max, α)
- Aperture a[i] = γ × l[i]^D
              - Random position (x,y) and orientation θ
           SAVE fracture geometries to frac_0_r.txt

     2. Grid Discretization (SubDivideToGrid.m):
FOR each fracture f in DFN:
           FOR each grid block (i,j) where i∈[1,nx], j∈[1,ny]:
              IF fracture f intersects grid block (i,j):
                 Clip fracture to grid boundaries
                 Adjust corner points by move_ratio to avoid meshing issues
STORE clipped segment in resu_0_r.txt with:
                    - Grid ID = i + (j-1)×nx
                    - Endpoints (x1,y1,x2,y2)
                    - Aperture w

3. DFM Simulation Setup (DiscreteFractureInput.m):
        FOR each grid block g with fractures:
           Create MRST input file s[g].m:
              - Define unstructured triangular mesh
              - Embed fractures as lower-dimensional elements
- Set matrix permeability = k_matrix
              - Set fracture permeability = w²/12

     4. Flow Simulation (GridDFMSimu.m):
        FOR each grid block g:
FOR each pressure gradient direction d ∈ {x, y}:
              Apply linear boundary conditions:
                 IF d = x: p_left = L, p_right = 0, p_top/bottom = linear
```

```
            IF d = y: p_top = L, p_bottom = 0, p_left/right = linear
            Solve flow equation: ∇·(k∇p) = 0
SAVE boundary fluxes q[boundary,d]

    5. Equivalent Permeability Calculation (CalcuKeq.m):
       FOR each grid block g:
          FOR each boundary b ∈ {left, right, top, bottom}:
Calculate total flux:
                q_total[b] = Σ(q_matrix[b] + q_fracture[b]·n)
                where n = normal vector

          Compute K_eq components using Darcy's law:
From x-direction flow:
                K_xx = q_x,out × d / (Δp × A)
                K_yx = q_y,out × d / (Δp × A)
             From y-direction flow:
                K_xy = q_x,out × d / (Δp × A)
K_yy = q_y,out × d / (Δp × A)

          Validate and correct:
             IF K_ii ≤ 0 OR K_ii > threshold:
                K_ii = k_matrix
IF |K_ij| < k_matrix:
                K_ij = 0

          SAVE K_eq to epcomf.txt

END Algorithm
```

## A.2 Implementation Structure

**Table A1. Computational modules and their implementation**

| Module | Implementation File | Primary Function | Key Parameters |
|---|---|---|---|
| DFN Generation | DiscreteFractureModeling.m | Generate stochastic fracture networks | num_real, n, α, l_min, l_max |
| Grid Partitioning | SubDivideToGrid.m | Clip and assign fractures to grid blocks | nx, ny, move_ratio |
| Simulation Setup | DiscreteFractureInput.m | Create MRST-compatible DFM models | mesh_size, k_matrix |
| Flow Solver | GridDFMSimu.m | Execute batch flow simulations | solver_type (TPFA/MPFA) |
| Upscaling | CalcuKeq.m | Calculate equivalent permeability tensors | tolerance, validation_threshold |

**Table A2. Data structure specifications**

| Data Type | Format | Structure | Description |
|---|---|---|---|
| Fracture geometry | ASCII text | 5 columns: x1, y1, x2, y2, aperture | Initial fracture network definition |
| Grid-fracture mapping | ASCII text | grid_ID, fracture segments | Fractures assigned to each grid block |
| Flux output | ASCII text | boundary_ID, flux_x, flux_y | Computed boundary fluxes |
| Permeability tensor | ASCII text | kxx, kxy, kyx, kyy | Final upscaled properties |