# Peer review of "UpsFrac v1.0: An open-source software for integrating modelling and upscaling permeability for fractured porous rocks"

_EGUsphere, 2025_

## Referee Comment (RC1)

The manuscript presents UpsFrac, a MATLAB-based workflow that links (i) generation of deterministic and stochastic discrete-fracture models (DFM) and (ii) flow-based upscaling of their equivalent permeability using the Multiple-Boundary Method (MBM) implemented with TPFA/MPFA discretization. The authors provide validation against analytical solutions, a field-scale demonstration, and release the code under GPL-3.0 on GitHub and Zenodo. The topic is timely; few open-source tools bridge fracture modelling and rigorous flow upscaling in a single framework. The paper would be publishable after a major revision that addresses methodological clarifications, strengthens validation, and improves presentation.

**General Comments**

1. While UpsFrac provides a useful integrated workflow, the individual components (ADFNE, MRST, MBM) are not novel. The paper would benefit from a clearer positioning of what is truly new: Is it the integration? Is it the application to fractal systems? Is it the ability to handle both deterministic and stochastic fractures? In the introduction section, clearly delineate the innovation over existing tools like PorePy, DFNWorks, or OpenGeoSys-DFM modules.

2. The accuracy of upscaling strongly depends on mesh size, particularly for fractures aligned with or against the grid. Include a sensitivity analysis showing how grid refinement or fracture-matrix contrast influences permeability estimates. Report convergence trends and potential sources of numerical error.

3. The code assumes constant aperture per fracture and a simplified grid-based representation. Discuss the implications of this assumption for faults with variable aperture or roughness. It would be better to quantify the error induced by this simplification. Consider adding a test where a fracture with log-normal aperture variability is upscaled with (i) constant-aperture assumption and (ii) a higher-fidelity solver (e.g., locally refined mesh or transfer-function approach)

4. The current implementation is 2D, which is acceptable for early-stage validation and pedagogy but limits industrial applicability. Provide a roadmap or discussion of how 3D capabilities could be implemented.

5. Computational performance and scalability are not analyzed. Provide runtime and memory statistics for the field-scale case (number of grids and fractures, CPU/GPU specs). Compare

with at least one alternative open-source package (e.g., OGS) to highlight UpsFrac's efficiency or identify bottlenecks.

6. Two validation tests (power-law DFN and single fracture with variable aperture) are useful but narrow. Add a heterogeneous matrix-fracture system where analytical REV permeability is known. Then compare it with embedded-discrete-fracture simulation on the full domain to confirm that the upscaled EFM reproduces domain-scale responses

7. The abstract claims that UpsFrac can easily run DFM ensembles for uncertainty analysis, yet no ensemble results are shown. Include an example with 50–100 stochastic realisations, report statistics (mean, variance) of the upscaled tensor components, and discuss convergence

8. Algorithm description lacks reproducibility details. Scripts such as "SubDivideToGrid.m" or "CalcuKeq.m" are mentioned but not summarized. Add a concise pseudo-code or flowchart (in the paper) that maps each processing step to its script name; list key input parameters

9. Frequent grammatical errors and misspellings distract from the content. Thorough language editing is required; see minor comments below for examples.

**Technical corrections**

1. In the abstract, clarify that the current implementation is 2-D and MATLAB-based.
2. Fig. 2 workflow: fonts are blurry; consider vector graphics.
3. Fig. 7 permeability map: add color-bar units
4. Fig. 9 ellipses: scale key missing (are axes normalized?).
5. Consistency in references: some entries list full journal names, others abbreviations; unify style.

---

## Author Response (AR1)

**Author's response**

**Reviewer 1**

The manuscript presents UpsFrac, a MATLAB-based workflow that links (i) generation of deterministic and stochastic discrete-fracture models (DFM) and (ii) flow-based upscaling of their equivalent permeability using the Multiple-Boundary Method (MBM) implemented with TPFA/MPFA discretization. The authors provide validation against analytical solutions, a field-scale demonstration, and release the code under GPL-3.0 on GitHub and Zenodo. The topic is timely; few open-source tools bridge fracture modelling and rigorous flow upscaling in a single framework. The paper would be publishable after a major revision that addresses methodological clarifications, strengthens validation, and improves presentation.

**Response to comments:**

We sincerely appreciate your thorough and constructive review of our manuscript on UpsFrac. Your detailed comments will significantly improve the quality and impact of our work. We are committed to addressing all your concerns comprehensively. We have carefully considered all comments and already initiated several improvements. Below, we provide our initial response to each point and our planned revisions.

**Comment 1:** While UpsFrac provides a useful integrated workflow, the individual components (ADFNE, MRST, MBM) are not novel. The paper would benefit from a clearer positioning of what is truly new: Is it the integration? Is it the application to fractal systems? Is it the ability to handle both deterministic and stochastic fractures? In the introduction section, clearly delineate the innovation over existing tools like PorePy, DFNWorks, or OpenGeoSys-DFM modules.

**Response to comments:** We appreciate the referee's recognition of UpsFrac as a "useful integrated workflow" and the constructive suggestion to better articulate

our innovations. We agree that clearer positioning will strengthen the manuscript. While the individual components (ADFNE, MRST, MBM) are indeed established methods, UpsFrac's innovation lies in their seamless integration and extension. Specifically, our contributions include: (1) The first open-source platform that directly bridges fractal theory with practical permeability upscaling through automated workflow—eliminating intermediate file conversions that cause data loss and computational overhead; (2) Unified handling of both deterministic and stochastic fractal fracture networks with power-law distributions and physically-based aperture-length correlations within a single framework; (3) Automated ensemble-based uncertainty quantification that reduces analysis time from days to hours, making large-scale Monte Carlo analysis accessible; (4) Extensible modular architecture enabling easy incorporation of new algorithms and adaptation to specific geological scenarios.

**Changes in manuscript:** We have revised the Introduction (last two paragraphs) to clearly position UpsFrac's innovations and added Table 1 for systematic comparison with existing tools. The positioning statement now explicitly states our unique contribution: bridging the gap between fractal DFN generation and flow-based permeability upscaling within an integrated framework that enables automated uncertainty quantification—a capability combination currently unavailable in any existing open-source platform.

**Comment 2:** The accuracy of upscaling strongly depends on mesh size, particularly for fractures aligned with or against the grid. Include a sensitivity analysis showing how grid refinement or fracture-matrix contrast influences permeability estimates. Report convergence trends and potential sources of numerical error.

**Response to comments:** We fully agree with this excellent point. Grid refinement effects, fracture-matrix permeability contrast, and fracture orientation relative to grid alignment are indeed crucial factors affecting upscaling accuracy.

We recognize that these parameters can significantly influence the computed equivalent permeability tensor and appreciate the reviewer highlighting this important aspect.

**Changes in manuscript:** We have added a new Section 3.3 (Numerical Accuracy and Sensitivity Analysis) that comprehensively evaluates UpsFrac's computational reliability. This section presents: (1) Grid convergence studies using six progressively refined meshes (2×2 to 80×80), demonstrating excellent convergence with errors below 1% for all tested resolutions and confirming second-order accuracy through Richardson extrapolation; (2) Fracture-matrix permeability contrast analysis spanning four orders of magnitude ($10^2$ to $10^6$), showing robust performance with mean error of only 3.40% and maximum error of 5.11%; (3) Orientation-dependent error quantification for fractures at 0°, 30°, 45°, 60°, and 90°, revealing mean component errors of 1.10-3.15% with maximum error of 7.67% at 45° due to corner node effects. These systematic investigations establish rigorous bounds on numerical accuracy and confirm UpsFrac's robustness across diverse fracture configurations.

**Comment 3:** The code assumes constant aperture per fracture and a simplified grid-based representation. Discuss the implications of this assumption for faults with variable aperture or roughness. It would be better to quantify the error induced by this simplification. Consider adding a test where a fracture with log-normal aperture variability is upscaled with (i) constant-aperture assumption and (ii) a higher-fidelity solver (e.g., locally refined mesh or transfer-function approach)

**Response to comments:** We thank the reviewer for this valuable suggestion. The constant aperture assumption indeed represents a significant simplification widely adopted in discrete fracture modeling due to computational efficiency and limited field data availability. Following your recommendation, we have conducted a comprehensive quantitative analysis comparing upscaled permeabilities

computed using constant versus variable aperture distributions, with lognormal variability representing realistic fracture roughness conditions.

**Changes in manuscript:** We have added Section 5.1 providing: (1) Systematic error quantification using fractures with coefficients of variation (CV) ranging from 0.1 to 0.5, representing mild to severe aperture heterogeneity; (2) Direct comparison of equivalent permeabilities between constant and variable aperture models; (3) Clear delineation of applicability bounds showing the constant aperture assumption remains valid (error <5%) for CV < 0.2 but becomes inadequate for rough fractures with CV > 0.3; (4) Implementation roadmap for incorporating variable aperture capabilities in future UpsFrac releases.

**Comment 4:** The current implementation is 2D, which is acceptable for early-stage validation and pedagogy but limits industrial applicability. Provide a roadmap or discussion of how 3D capabilities could be implemented.

**Response to comments:** We agree that while the current 2D implementation is well-suited for validation and pedagogy, 3D capability is essential for industrial applicability. We have previously demonstrated 3D upscaling (Chen et al., 2018, Hydrogeology Journal 26:1903-1916), providing a proven methodological foundation that has not yet been integrated into UpsFrac. Building on our existing DFN generation and simulation tooling (e.g., ADFNE and MRST), the 3D extension involves three key technical challenges: geometric complexity of fracture intersections, computational scaling, and memory management.

**Changes in manuscript:** We have added a comprehensive Section 5.2 "Three-dimensional model extension roadmap" that outlines concrete acceleration strategies and a development roadmap covering: (i) generalizing DFN generation from 2D line fractures to 3D planar polygonal or triangulated fracture surfaces with realistic size, orientation, and aperture distributions; (ii) extending the multiple-boundary fracture upscaling (MFU) method, also known as

the multiple boundary method (MBM), to 3D with robust treatment of fracture intersections and complex connectivity patterns; and (iii) mitigating the substantially higher computational cost through algorithmic accelerations, including adaptive/octree meshing, hierarchical fracture-surface meshing, algebraic multigrid preconditioning, domain decomposition with parallel CPU/GPU solvers, and graph-based pruning of disconnected subnetworks. We anticipate that while technically feasible, the primary challenges will be twofold: managing the cubic growth in computational requirements (where transforming a 100×100 2D grid into 100×100×100 in 3D increases the problem size from 10,000 to 1 million nodes) and ensuring robust handling of complex 3D fracture intersection geometries.

**Comment 5:** Computational performance and scalability are not analyzed. Provide runtime and memory statistics for the field-scale case (number of grids and fractures, CPU/GPU specs). Compare with at least one alternative open-source package (e.g., OGS) to highlight UpsFrac's efficiency or identify bottlenecks.

**Response to comments:** We thank the reviewer for highlighting the importance of computational performance analysis. We have now added a comprehensive performance evaluation in the revised manuscript.

**Changes in manuscript:** We have added Section 4.2 that presents a detailed computational performance analysis for a field-scale fractured geothermal system. The analysis was conducted on an Intel Core i5-9300H CPU (2.40 GHz) with 8 GB RAM running Windows 10. The upscaling workflow consists of five computational stages with corresponding runtime breakdown: (1) Fracture-Coarse Grid Partitioning stage (0.186s) assigns fractures to Cartesian blocks using geometric intersection algorithms; (2) Simulation Setup stage (28.659s) creates the DFM using TPFA/MPFA discretization and cubic law; (3) Flow Simulation stage (71.245s) computes directional flux fields by solving Darcy's law with orthogonal pressure

gradients; and (4) Upscaling Calculation stage (49.650s) applies the multiple boundary approach (Eq. 3) to calculate the equivalent permeability tensor. The total upscaling runtime is 149.74s, plus 540s for flow simulation, yielding 689.74s total. Direct benchmarking against OpenGeoSys (OGS) demonstrates significant efficiency gains, with our approach achieving a 67% total runtime reduction (689.74 s vs 2102.45 s) and 89% total memory reduction (24.33 MB vs 214.71 MB), as detailed in Figure 18.

**Comment 6:** Two validation tests (power-law DFN and single fracture with variable aperture) are useful but narrow. Add a heterogeneous matrix-fracture system where analytical REV permeability is known. Then compare it with embedded-discrete-fracture simulation on the full domain to confirm that the upscaled EFM reproduces domain-scale responses

**Response to comments:** e appreciate the reviewer's suggestion to include broader validation with heterogeneous matrix-fracture systems. We have now incorporated a comprehensive field-scale validation case.

**Changes in manuscript:** Section 4.2 now includes a comprehensive field-scale validation using a 500m × 500m fractured geothermal reservoir. We present a direct comparison between the upscaled EFM model (400 cells) and a high-fidelity DFM simulation (13,414 nodes), with temperature breakthrough curves over 30 years showing excellent agreement—achieving a mean relative error below 0.153% and median error of only 0.036%. The model incorporates realistic heterogeneous fracture-matrix coupling with a fracture aperture of 500 μm and matrix permeability of $9.87 \times 10^{-16}$ m², validating the method's ability to handle complex heterogeneous systems. Despite an 84% reduction in grid cells, the upscaled model accurately reproduces the domain-scale thermal-hydraulic behavior throughout the entire simulation period. This comprehensive validation confirms that our upscaling approach reliably captures the complex physics of

heterogeneous fractured systems while achieving significant computational efficiency.

**Comment 7:** The abstract claims that UpsFrac can easily run DFM ensembles for uncertainty analysis, yet no ensemble results are shown. Include an example with 50–100 stochastic realisations, report statistics (mean, variance) of the upscaled tensor components, and discuss convergence.

**Response to comments:** We thank the reviewer for this valuable suggestion regarding uncertainty quantification. We agree that demonstrating ensemble-based uncertainty quantification is essential for validating the abstract's claims, especially given the inherent uncertainty in subsurface observations due to data sparsity and noise.

**Changes in manuscript:** We have added a comprehensive ensemble uncertainty quantification analysis in Section 4.3 featuring 100 stochastic DFN realizations. This new section includes: (1) Complete statistical analysis of upscaled permeability tensor components ($k_{xx}$, $k_{yy}$, $k_{xy}$) with means, standard deviations, and coefficients of variation; (2) Distribution characterization showing log-normal behavior for diagonal components and normal distribution for off-diagonal components, confirmed through Q-Q plot analysis; (3) Convergence assessment demonstrating that ensemble statistics stabilize after 50-60 realizations with relative errors below 5%; (4) Computational performance metrics showing 108 minutes total runtime for 100 realizations with 8.2 GB peak memory usage; (5) Identification of flow simulation as the primary computational bottleneck (65% of runtime). The results validate UpsFrac's capability for Monte Carlo workflows and demonstrate how the tool transforms complex discrete fracture systems into stochastic continuum representations while preserving critical uncertainty information for subsurface risk assessment.

**Comment 8:** Algorithm description lacks reproducibility details. Scripts such as "SubDivideToGrid.m" or "CalcuKeq.m" are mentioned but not summarized. Add a concise pseudo-code or flowchart (in the paper) that maps each processing step to its script name; list key input parameters

**Response to comments:** We thank the reviewer for this valuable suggestion regarding algorithm reproducibility. We agree that the current algorithm description lacks sufficient detail and intuitive presentation, and that the functions of key scripts are not adequately summarized. This indeed affects the understanding and usability of the software algorithm.

**Changes in manuscript:** We have enhanced algorithm reproducibility through three complementary additions: (1) Revised Figure 2 as a comprehensive implementation flowchart; (2) Restructured Table 2 in Section 2.3 to emphasize mathematical frameworks and computational stages; (3) Expanded Appendix A with two sections: A.1 providing detailed pseudocode for the complete upscaling workflow, and A.2 containing script mapping and file specifications.

**Comment 9:** Frequent grammatical errors and misspellings distract from the content. Thorough language editing is required; see minor comments below for examples.

**Response to comments:** We thank the reviewer for pointing out the language issues and acknowledge that grammatical errors and misspellings indeed distract from the technical content. We apologize for these oversights and agree that thorough language editing is essential for clear scientific communication.

**Changes in manuscript:** We have conducted comprehensive language editing throughout the revised manuscript: (1) Corrected all grammatical errors including article usage ("a integrated"→"an integrated"), verb forms ("in proposed"→"is proposed", "should defined"→"should be defined"), and word choice ("determinedly"→"deterministically"); (2) Fixed all typographical errors and

misspellings throughout the text; (3) Removed redundant passages and improved sentence structure for clarity; (4) Standardized technical terminology and mathematical notation; (5) Ensured consistent formatting of references and citations.

**Technical corrections 1:** In the abstract, clarify that the current implementation is 2-D and MATLAB-based.

**Response to comments:** Thank you for this valuable suggestion. We have clarified the technical implementation details in the abstract as requested.

**Changes in manuscript:** We have added the following clarification to the abstract: "The current implementation is 2D and MATLAB-based, built upon fracture modelling code ADFNE and reservoir simulation code MRST, which can easily run DFM ensembles for uncertainty analysis."

**Technical corrections 2:** Fig. 2 workflow: fonts are blurry; consider vector graphics.

**Response to comments:** Thank you for pointing out the font quality issue in Figure 2. We agree that clear, high-quality graphics are essential for effective communication of the workflow.

**Changes in manuscript:** We have recreated Figure 2 using vector graphics to ensure crisp, clear fonts and improved overall visual quality throughout the workflow diagram.

**Technical corrections 3:**

**Response to comments:** Thank you for this important suggestion to improve the clarity of the permeability visualization.

**Changes in manuscript:** We have added colorbar units to the permeability map in Figure 7 (now Figure 12 in the revised manuscript due to additional figures).

Similarly, we have checked Figure 8 (now Figure 13 in the revised manuscript) and added appropriate units as well.

**Technical corrections 4:** Fig. 9 ellipses: scale key missing (are axes normalized?).

**Response to comments:** Thank you for pointing out the missing scale information for the ellipse visualization. You are correct that the axes are normalized. The normalization is intentional to better visualize anisotropy directions across the domain, as the permeability values vary significantly.

**Changes in manuscript:** We have clarified the scaling approach in Figure 9 (now Figure 14 in the revised manuscript) by updating the figure caption to specify that "Each ellipse is normalized within its subplot to enhance visualization of tensor orientation rather than absolute permeability magnitude."

**Technical corrections 5:** Consistency in references: some entries list full journal names, others abbreviations; unify style.

**Response to comments:** Thank you for pointing out the inconsistency in journal name formatting throughout our reference list.

**Changes in manuscript:** We have checked and standardized all journal names to use their full titles rather than abbreviations for consistency.  All references now consistently use complete journal names throughout the bibliography.

**Reviewer 2**

The paper introduces UpsFrac v1.0, an open-source MATLAB-based tool for fracture modelling and permeability upscaling. While the integration of fracture network generation, flow simulation, and upscaling into one workflow has potential value, the manuscript in its current form requires major revision.

**Response to comments:**

We sincerely appreciate your thorough and constructive review of our manuscript. Your detailed feedback will significantly improve the clarity and scientific contribution of our work. We agree with your assessment that the manuscript requires major revision, and we are committed to addressing all your concerns comprehensively. Below, we provide our point-by-point response and planned revisions.

**Comment 1:** The paper focuses on scripts rather than science. Much of the text reads like a user manual (e.g., Section 2, lines 135–137 and 140–146), with repeated references to scripts, file names, and formats. These details should be moved to the GitHub documentation. The manuscript should instead emphasize scientific contribution and comparison with existing approaches. Throughout the paper, please remove file names and input/output formats and focus on the science. For example:

Lines 113–114: "The deterministic fracture can be stored in the file frac_deterministic.txt. Each line contains geometric data of a fracture." → This sentence can be removed.

Lines 184–190: Instead of describing what scripts to use, this could be rewritten as:

"Using the upscaling results, the distribution of permeability tensor components (kxx, kxy, kyy) can be visualized to characterize spatial heterogeneity and

anisotropy. In addition, the equivalent permeability tensor of each grid block can also be plotted as an ellipse to visualize the directional permeability."

**Response to comments:** We sincerely thank the reviewer for this critical feedback. We fully agree that the manuscript read too much like a user manual rather than focusing on scientific contributions. We have comprehensively revised Section 2 and throughout the manuscript to emphasize the mathematical formulations, algorithmic innovations, and scientific methodology.

**Changes in manuscript:** We have addressed this issue through: (1) Removed all file names, format specifications, and script references from the main text throughout Section 2; (2) Restructured Table 2 to emphasize computational stages and mathematical formulations (power-law distributions, TPFA/MPFA schemes, cubic law, multi-boundary methods) rather than script names; (3) Revised Sections 2.2-2.3 to focus on scientific methodology and algorithmic innovations; (4) Relocated all detailed file formats, script names, and implementation specifications to Appendix A Section A.2, while keeping the main text focused on scientific methodology; (5) Enhanced the visualization section following the reviewer's specific suggestion to emphasize characterization of spatial heterogeneity and anisotropy through permeability tensor analysis; (6) Additionally, we have thoroughly reviewed all scripts in the GitHub repository and added comprehensive comments for parameters throughout the codebase to improve usability and help users better understand the implementation of our methodologies.

**Comment 2:** The manuscript requires substantial editing for language, terminology, abbreviations, and typos.

- Many sentences are difficult to follow. For example, lines 163–165: "For each grid, two simulations are conducted that the pressure gradient along the x and y-axis. The results used for upscaling are saved, including the flux, the centre of

the face, and the index of the grid face (fracture or matrix)" is grammatically incorrect and confusing. It should explain more clearly that two separate simulations are performed. A clearer version would be:

"For each grid, two separate simulations are performed: one with a pressure gradient applied along the x-axis and one with a gradient along the y-axis. From these simulations, fluxes, the centre of the face, and the index of the grid face are computed for the upscaling step."

- Some sentences are confusing. For example: "long/short tail as alpha is high" (line 198), or "Different components of equivalent permeability are different, which is influenced by the fracture orientation" (lines 243–244). This should be corrected throughout the manuscript.

- There are several grammatical issues throughout the text. Careful editing is required.

- Abbreviations are not handled properly. Once terms such as DFM or EFM are introduced, the full forms are repeated unnecessarily (see lines 41–46). Please revise throughout the manuscript.

- Some terms are inconsistent or unclear. For example: "engineering structures" (line 27) is ambiguous and out of context, and "MFU" (line 17) is never defined (what does it stand for?). Please revise for clarity and consistency.

-Typographical errors: Examples include "aline" → "aligned" and "natual" → "natural". Please proofread carefully throughout the text. Please also check that a correct DOI is provided (line 310).

**Response to comments:** We acknowledge the numerous language issues and apologize for these oversights that detract from the technical content. We greatly appreciate your detailed suggestions for improvement.

**Changes in manuscript:** We have systematically addressed all language issues: (1) Corrected grammatical errors throughout, including the confusing sentence structure in lines 163-165, now clearly stating that simulations are conducted with pressure gradients applied along both x- and y-axes; (2) Improved unclear expressions such as "long/short tail as alpha is high" (line 198) to "heavy tail for low α values and light tail for high α values"; (3) Restructured redundant sentences like lines 243-244 from "Different components...are different" to "The anisotropy in equivalent permeability components is controlled by the fracture orientation distribution"; (4) Standardized abbreviation usage - DFM and EFM are now defined once and used consistently throughout; (5) Clarified ambiguous terms by replacing "engineering structures" with "geological formations"; (6) Properly defined MFU as "multiple-boundary fracture upscaling" in the abstract; (7) Corrected all typographical errors including "Limitions"→"Limitations", "Fractrue"→"Fracture", "natual"→"natural", and "permeably"→"permeability"; (8) Verified the Zenodo DOI (https://doi.org/10.5281/zenodo.14674083) is correct and accessible.

**Comment 3:** In terms of scientific novelty, the main contribution appears to be the implementation of the multiple boundary method to calculate equivalent permeabilities. However, this method is not clearly explained, and MRST already provides equivalent permeability upscaling for fracture networks (see Lie & Møyner, Advanced Modeling with the MATLAB Reservoir Simulation Toolbox, 2021, Chapter 9.4). The authors should clarify why they did not use MRST's existing routines for calculating the upscaled permeability of the grids, and what advantages their approach offers. Furthermore, the use of a power-law distribution for fracture lengths is presented as a novelty, but this is not new. Most DFN generators, including several cited in the introduction, already support this capability. It should therefore not be highlighted as a main innovation.

**Response to comments:** We thank the reviewer for this important clarification request. We acknowledge that power-law distributions are indeed standard in DFN

generators—we will revise to avoid presenting this as novel. Regarding the Multiple Boundary Method (MBM, also referred to as MFU in some literature) and comparison with MRST: While MRST (Chapter 9, Lie & Møyner, 2021) provides upscaling through embedded discrete fracture models (EDFM), UpsFrac employs a fundamentally different approach. MRST's EDFM uses non-conforming grids where fractures are embedded in matrix cells, suitable for large-scale models but with limitations in accuracy for highly connected fracture networks. In contrast, UpsFrac uses conforming DFM grids with TPFA/MPFA schemes and MBM upscaling (Chen et al., 2015), which: (1) Provides higher accuracy for complex fracture intersections through explicit geometric representation; (2) Enables flexible boundary condition assignment on multiple domain faces (both fracture and matrix boundaries), crucial for accurate permeability tensor calculation; (3) Better captures local flow behavior in densely fractured zones where EDFM approximations may lose accuracy. Our innovation is not MBM itself, but its integration with fractal DFN generation in an automated workflow enabling ensemble-based uncertainty quantification.

**Changes in manuscript:** We have: (1) Removed claims about power-law distributions being novel; (2) Added explicit comparison with MRST's EDFM approach (Lie & Møyner, 2021, Ch. 9), highlighting the technical differences and specific scenarios where our DFM-MBM approach offers advantages; (3) Clarified that our main innovation is the integrated workflow that automates the entire process from fractal DFN generation to upscaled properties with uncertainty quantification, not the individual methods themselves.

**Comment 4:** The validation section (lines 205–215) is very unclear. It is not obvious what simulation setup was used. Was it a single fracture inside a low-permeability matrix as shown in Fig. 5, or a single isolated fracture? This needs to be clarified. In addition, the manuscript refers to an "analytical solution" but does not provide the actual equation. Without this information, the validation

is almost impossible to follow and not reproducible in its current form. This section should be rewritten to clearly describe the geometry and the simulation setup, the analytical solution used, and the assumptions behind the comparison.

**Response to comments:** We thank the reviewer for this constructive comment. We agree that the validation section lacks essential details for clarity and reproducibility. We have now: (1) Clarified that the simulation involves a single fracture embedded in a low-permeability matrix with explicit boundary conditions, analytical solution equations (Section 3.2), and underlying assumptions for the comparison; (2) Added a comprehensive Section 3.3 "Numerical Accuracy and Sensitivity Analysis" containing grid convergence studies, permeability contrast analysis ($10^2$-$10^6$ range), and orientation-dependent error assessment (0°-90°), demonstrating mean relative errors below 5% across diverse conditions.

**Changes in manuscript:** We have completely rewritten Section 3.2 with detailed simulation setup, analytical solution equations, and explicit assumptions. Additionally, as detailed in our response to Reviewer 1 Comment 2, we added Section 3.3 "Numerical Accuracy and Sensitivity Analysis" with three validation studies: (i) Grid convergence (3.3.1) showing <0.65% error even for coarse grids; (ii) Permeability contrast sensitivity (3.3.2) validating accuracy across four orders of magnitude ($10^2$-$10^6$) with 3.40% mean error; and (iii) Orientation analysis (3.3.3) confirming accurate tensor computation for 0°-90° fracture angles with mean errors of 1.10% ($K_{xx}$), 3.15% ($K_{yy}$), and 0.07% ($K_{xy}$). These additions directly address the reviewer's concerns about validation clarity and reproducibility.

**Comment 5:** Nowhere in the manuscript is performance discussed. There is no information on computation time, memory requirements, or applicability to larger networks, yet these are critical aspects for evaluating an upscaling tool. Please include an assessment of computational performance in the revised version.

**Response to comments:** We thank the reviewer for this valuable suggestion and acknowledge that computational performance analysis was indeed missing from our original manuscript. We fully agree that these metrics are critical for assessing the practical utility for field-scale applications.

**Changes in manuscript:** As detailed in our response to Reviewer 1 Comment 5, we have added Section 4.2 reporting performance of the upscaling method, including: (1) Detailed performance metrics for field-scale fracture networks; (2) Wall-clock runtime and memory usage for key computational stages; (3) Direct comparison with open-source DFM simulators OpenGeoSys (OGS) and SHEMAT using identical inputs and hardware; (4) Complete hardware specifications for reproducible benchmarking (CPU model, RAM, OS); (5) Results summarized in Figure 18 demonstrating UpsFrac's 67% runtime reduction and 89% memory savings. Additionally, section 4.3 reports ensemble performance for 100 stochastic realizations, requiring 108 minutes total runtime with 8.2 GB peak memory usage.